# Calving front positions for 42 key glaciers of the Antarctic Peninsula Ice Sheet: a sub-seasonal record from 2013 to 2023 based on a deep learning application to Landsat multispectral imagery

Erik Loebel[1,2], Celia A. Baumhoer[3], Andreas Dietz[3], Mirko Scheinert[1], and Martin Horwath[1]

[1]Technische Universität Dresden, Institut für Planetare Geodäsie, Dresden, Germany
[2]Alfred-Wegener-Institut Helmholtz Zentrum für Polar- und Meeresforschung, Sektion Glaziologie, Bremerhaven, Germany
[3]German Aerospace Center, Earth Observation Center, Weßling, Germany

**Correspondence:** Erik Loebel (erik.loebel@tu-dresden.de)

**Abstract.** Calving front positions of marine-terminating glaciers are an essential parameter for understanding dynamic glacier changes and constraining ice modelling. In particular, for the Antarctic Peninsula, where the current ice mass loss is driven by dynamic glacier changes, accurate and comprehensive data products are of major importance. Current calving front data products are limited in coverage and temporal resolution because they rely on manual delineation, which is time-consuming and unfeasible for the increasing amount of satellite data. To simplify the mapping of calving fronts we apply a deep learning based processing system designed to automatically delineate glacier fronts from multispectral Landsat imagery. The U-Net based framework was initially trained on 869 Greenland glacier front positions. For this study we extended the training data by 252 front positions of the Antarctic Peninsula. The data product presented here includes 4817 calving front locations of 42 key outlet glaciers from 2013 to 2023 and achieves sub-seasonal temporal resolution. The mean difference between automated and manual extraction is estimated at $59.3 \pm 5.9\,\mathrm{m}$. This dataset will help to better understand marine-terminating glacier dynamics on an intra-annual scale, study ice-ocean interactions in more detail and constrain glacier models. The data is publicly available at PANGAEA under https://doi.pangaea.de/10.1594/PANGAEA.963725 (Loebel et al., 2023).

## 1 Introduction

From 1992 to 2020 the Antarctic Ice Sheet lost $2671 \pm 530\,\mathrm{Gt}$ of ice, raising the global sea level by $7.4 \pm 1.5\,\mathrm{mm}$ (Otosaka et al., 2023). Mass loss is dominated by ice-dynamic processes, where a decrease of ice shelf thickness and extent reduces buttressing and thereby accelerates the ice flow discharge of grounded ice across the grounding line (Slater et al., 2020). At the Antarctic Peninsula (AP) in particular, increasing ice loss has been linked with ice shelf disintegration (Rott et al., 1996; Rignot et al., 2004; Rack and Rott, 2004; Cook and Vaughan, 2010). Thereby, atmospheric and oceanic influences cause ice shelf thinning and precondition collapse (Pritchard et al., 2012; Adusumilli et al., 2018). Although collapsing ice shelves do not directly contribute to sea level rise, they play an important role in stabilizing their tributary glaciers (Dupont and Alley, 2005). Once this support is lost, the dynamics of the tributary glaciers and their ice discharge increase, contributing directly to sea-level rise (Rignot et al., 2004; Seehaus et al., 2018). This mechanism has been observed in a number of cases most notably

after the collapse of the Prince Gustav ice shelf (Glasser et al., 2011) and the Larsen A and B ice shelves (Hulbe et al., 2008; Rott et al., 2011, 2018). Beyond that, even marine terminating glaciers that were not directly affected by ice shelf collapse, most of which flowing westward from the AP plateau, are experiencing changing dynamics as a result of warming climate (Cook et al., 2016; Hogg et al., 2017; Wallis et al., 2023a; Davison et al., 2024). The negative ice mass change rate of the entire AP, $-13 \pm 5\,\mathrm{Gt\,yr^{-1}}$ between 1992 and 2020 and $-21 \pm 12\,\mathrm{Gt\,yr^{-1}}$ between 2017 and 2020, represents 14 % (between 1992 and 2020) and 18 % (between 2017 and 2020) of the mass loss rate of the Antarctic Ice Sheet (Otosaka et al., 2023). Monitoring of AP marine terminating glaciers and ice shelves is of paramount importance for up-to-date diagnosis and reliable prediction of future changes.

One particularly important parameter of each glacier is the calving front position and its temporal variation. Firstly, calving front locations are the basis for mapping glacier area change. In this way, Cook et al. (2014) show that the majority of AP glaciers have reduced in area since the 1940s with temporal trends indicating uniform retreat since the 1970s. Most significant area losses occurred in the northeast AP and are associated with ice shelf collapse. Area loss on the west coast shows a north-south gradient and has been linked to warming ocean water (Cook et al., 2016). Secondly, calving front locations are essential for studying and understanding ice-ocean interaction as well as underlying processes. In this way, they help to understand the response of the AP to a warming climate. This applies to local studies of individual glaciers or glacier systems (Scambos et al., 2011; Seehaus et al., 2015, 2016), but also to regional studies (Friedl et al., 2018; Wallis et al., 2023a; Ochwat et al., 2024; Surawy-Stepney et al., 2024). Thirdly, calving front locations play an important role in constraining ice-dynamic models to improve simulations of future mass loss and sea level contribution (Alley et al., 2005; Barrand et al., 2013; Cornford et al., 2015). According to Pattyn and Morlighem (2020), calving is one of the key physical processes where its lack of knowledge reduces the ability to accurately predict mass changes of the Antarctic Ice Sheet and define potential tipping points. Modelling studies for Jakobshavn Isbræ in Greenland identified the calving as the dominant control with calving front migration accounting for 90 % of the acceleration (Bondizo et al., 2017). Similar conclusions are also drawn by Vieli and Nick (2011) who emphasise the need for a robust representation of the calving in ice sheet models.

Accurate glacier calving front data with both high temporal resolution and a high spatial coverage is therefore critical. At present, however, these data products are not widely available for the AP. This is due to limitations of the manual and therefore time-consuming, process of delineating these frontal positions from the increasing amount of satellite imagery available. Table 1 gives an overview of publicly available outlet glacier calving front datasets for the AP. The Antarctic Digital Database (ADD) (Cook et al., 2021b) and Global Land Ice Measurements from Space (GLIMS) (GLIMS Consortium, 2005; Raup et al., 2007) products have circum-Antarctic coverage but very limited temporal resolution. The calving front data by Seehaus et al. (2015), Seehaus et al. (2016), Wallis et al. (2023b) and Surawy-Stepney (2024) are by-products of glaciological studies. Calving fronts reported by Gourmelon et al. (2022) are part of a benchmark dataset developed for evaluating automated extraction from SAR imagery. For the vast majority of the approximately 800 marine-terminating glaciers at the AP (Cook et al., 2014; Huber et al., 2017), current data products do not exploit the potential of available satellite observations. The availability of glacier calving front positions at the AP is limited, emphasising the necessity for additional and more comprehensive data products. To generate these data products efficiently, we need to use automatic annotation methods.

**Table 1.** Overview of publicly available glacier calving front datasets for the AP Ice Sheet. The number of fronts mapped by Cook et al. (2021a) is not documented. It is specified that more than 2000 aerial photographs and over 100 satellite images were used to compile the dataset. The datasets listed are those that include AP outlet glacier calving fronts. Data products focusing on ice shelves (e. g. Greene et al., 2022; Baumhoer et al., 2023; Andreasen et al., 2023) are not listed here.

| dataset | Annotation | Sensor type | Glaciers | Mapped fronts | Time span |
|---|---|---|---|---|---|
| ADD (Cook et al., 2021a) | Manually | Optical | 244 | | 1843-2008 |
| GLIMS (GLIMS Consortium, 2005) | Manually | Optical | >300 | >900 | Since 1986 |
| Seehaus et al. (2015) | Manually | SAR | 1 | 147 | 1992-2014 |
| Seehaus et al. (2016) | Manually | SAR | 1 | 133 | 1993-2014 |
| CryoPortal (ENVEO) | Manually | SAR & Optical | 16 | 124 | 2013-2017 |
| Gourmelon et al. (2022) | Manually | SAR | 5 | 457 | 1996-2020 |
| Wallis et al. (2023b) | Manually | SAR | 8 | 3430 | 2015-2021 |
| Surawy-Stepney (2024) | Manually | SAR & Optical | 9 | 245 | 2002-2023 |
| This study (Loebel et al., 2023) | Automatic | Optical | 42 | 4817 | 2013-2023 |

In recent years, deep learning has emerged as the tool of choice to accomplish this task (Mohajerani et al., 2019; Baumhoer et al., 2019; Zhang et al., 2021; Heidler et al., 2021; Marochov et al., 2021; Periyasamy et al., 2022; Davari et al., 2022b, a; Heidler et al., 2022; Herrmann et al., 2023). This has already been demonstrated by Baumhoer et al. (2023), who applied neural networks on SAR imagery to generate a high temporal resolution dataset of Antarctic ice shelf frontal positions. On the AP this IceLines dataset (Baumhoer et al., 2023) solely encompasses the Larsen Ice Shelf and excludes the outlet glaciers. Similar methods have been used to generate calving front data products for outlet glaciers in Greenland (Cheng et al., 2021; Zhang et al., 2023; Loebel et al., 2024c) and Svalbard (Li et al., 2023).

With this contribution, we provide a dense calving front data product for 42 key glaciers of the AP. We achieve this by applying a processing system, initially developed for Greenland, and incorporating new reference data. The locations of these glaciers are shown in Figure 1. The period covered ranges from 2013 to 2023. Glaciers were chosen based on four criteria. We process all glaciers which are (1) part of the AP Ice Sheet, (2) marine-terminating, (3) listed in the SCAR Composite Gazetteer of Antarctica (Cervellati et al., 2000) and (4) have a minimum calving front length of 5 km. The first three criteria define the scope for this dataset, our product does not include glaciers on surrounding islands nor ice shelve tributaries, the fourth criterion is related to processing limitations.

## 2 Methods

The processing is based on the method previously described by Loebel et al. (2024c). Originally developed for marine-terminating outlet glaciers in Greenland, the method is built with a high degree of automatization. The main modification applied to the framework is the extension of the reference dataset to incorporate glaciers on the AP. Figure 2 gives a compre-

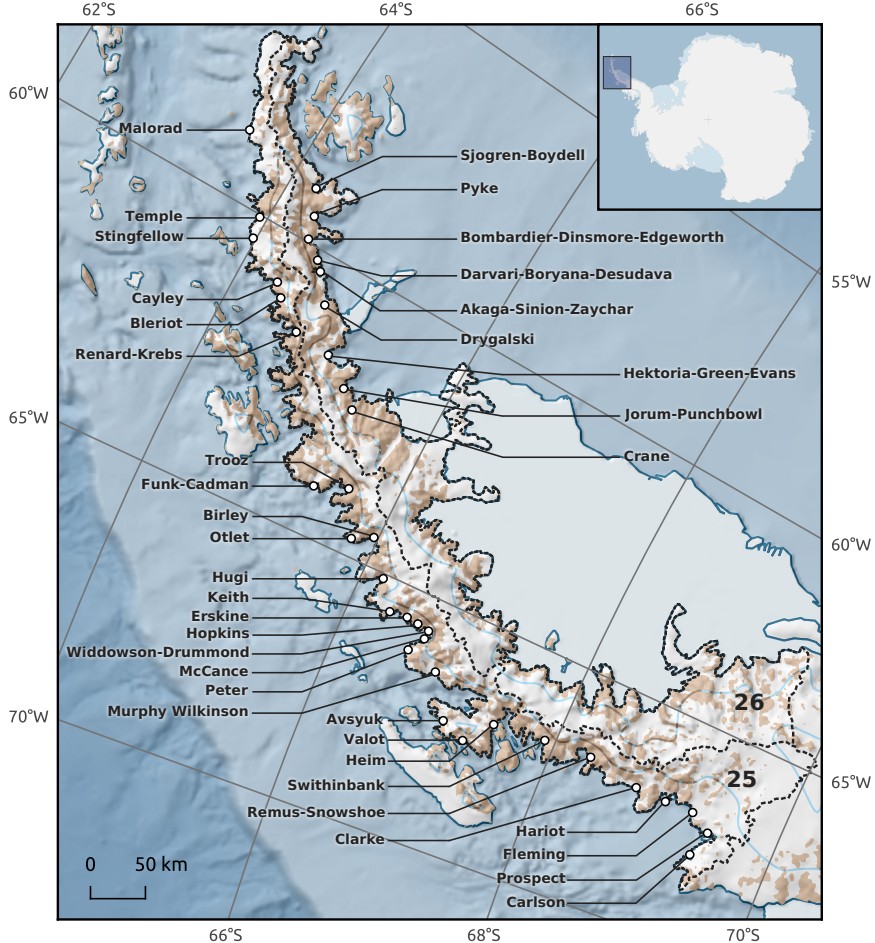

**Figure 1.** Overview map of the Antarctic Peninsula and the 42 glaciers included in the presented data product. All glaciers lie within the drainage basins 25 and 26 mapped by Zwally et al. (2012).

hensive overview of the processing system. The steps involved are described below, followed by an accuracy assessment of the results.

## 2.1 Calving front delineation using deep learning

Our processing is based on multispectral Landsat-8 and Landsat-9 Level-1 data. During pre-processing, nine available satellite bands, ranging from visible and infrared (VNIR) over short-wave infrared (SWIR) to thermal infrared (TIR), are cropped into $512\,\mathrm{px} \times 512\,\mathrm{px}$ tiles with a unified ground sampling distance of $30\,\mathrm{m}$, centered at the corresponding calving front. With a width of about $15.3\,\mathrm{km}$, these input tiles cover the calving front for most Greenland and AP outlet glaciers. To counteract image overexposure we apply a cumulative count cut image enhancement, clipping the data between the 0.1 and 98 percentile.

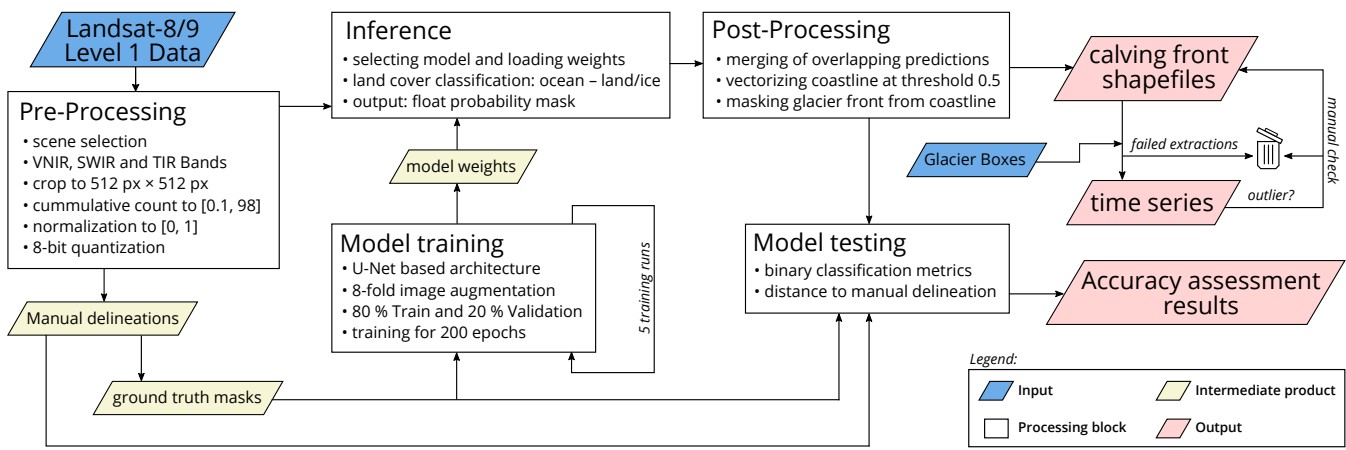

**Figure 2.** The workflow of the applied processing system divided into the various processing blocks.

Furthermore, all bands are normalized between 0 and 1 by a 8-bit quantization. Ground truth reference was inferred by manual
delineation for both training and testing our Artificial Neural Network (ANN). To train the model we apply 869 Greenland
calving front positions and additional 252 calving fronts from 12 AP glaciers. Due to the similar morphology of Greenland
and AP outlet glaciers, these 869 Greenland calving front positions represent an ideal basis for a well-generalized ANN model.
The additional 12 AP glaciers are Jorum, Punchbowl, Prospect, Hektoria-Green-Evans, Dryglaski, Birley, Crane, Widdowson,
Drummond, Fleming, Sjogren and Boydell. Expanding the training dataset is beneficial to account for the partly different
glacier morphology, such as the presence of free-floating glacier tongues. To avoid model overfitting, we make sure that the
training data covers different calving and ice mélange conditions, as well as varying illumination and cloud situations.

The applied ANN performs a land cover classification where an ocean class is semantically segmented from a glacier/land
class. In particular we use a modified U-Net (Ronneberger et al., 2015) with two additional contracting and expanding blocks.
This modification results in a larger receptive field, which is helpful for calving front extraction (Heidler et al., 2021). 20 % of
95 the input data is used for internal model validation and model selection. Training data is augmented eight times by rotation and
mirroring. For model training, we used the Adam optimization algorithm (Kingma and Ba, 2014) on a binary cross-entropy
loss function for 200 epochs and randomized batches of size eight. The model output is a floating point probability mask. Each
image pixel is assigned a probability between 0 (water) and 1 (glacier and land). Since the terminus length of the Hektoria-
Green-Evans glacier system exceeds the fixed window size, we infer five separate but partially overlapping predictions here.
We then merge these five predictions by averaging the values where they overlap. During post-processing the prediction is
vectorized using the Geospatial Data Abstraction Library (GDAL/OGR contributors, 2020) using a threshold of 0.5. The
threshold of 0.5 is the boundary between the predicted water and glacier/land classes, i.e. the predicted coastline. The glacier
front is then extracted by intersecting the predicted vectorized coastline with a static mask which is manually generated for
each glacier. Masked predictions therefore only contain the calving front. This is important not only to produce a consistent

data product, but also to perform a correct accuracy assessment, as the land-ocean boundary is almost static, making it easier for the ANN to delineate.

For further analysis, the calving front location shape-files are processed using the rectilinear box method (Moon and Joughin, 2008). We use this method not only to generate the time series of terminus area change but also to remove failed calving front extractions and separate outliers. In particular, calving fronts which do not split their corresponding glaciers box are discarded as failed extractions. In addition, we separate all entries that have an area difference of more than 20 % of the corresponding box width from the previous and following entries. Separated entries are checked manually and either (1) reinserted into the dataset if they were separated due to a true area change (e. g. due to calving of a large iceberg) or (2) discarded if the area change was due to a misclassification by the ANN.

For the generation of our data product we downloaded 4991 Landsat-8 and Landsat-9 Level-1 scenes, all available data for the 42 glaciers until May 2023. These are then preprocessed into 30453 stacked (9 bands, $512\,\mathrm{px} \times 512\,\mathrm{px}$) raster subsets. Since consecutive Level-1 Landsat scenes of the same path overlap, we select a maximum of one entry per day by minimising no-data pixels. The resulting 23230 raster subsets are processed by the ANN. More than half of the extractions fail, mostly due to cloud cover, leaving 8688 calving fronts. After outlier separation and checking, our final data product contains 4817 calving front positions. The success rate, which we define here as the ratio of raster subsets going into ANN processing to final quality controlled data product entries, is 21 %. All data product entries provided are full calving front extractions covering the entire calving front trajectory.

## 2.2 Accuracy assessment

The main accuracy assessment is done by comparing ANN-delineated calving front predictions to manual delineation for independent test imagery. The results of this comparison will validate our processing system and provide valuable metrics for comparing this method to existing studies. As an additional metric we introduce the inter-model distance. Although the inter-model distance has limited reliability, it has the advantage that it can be determined for each ANN prediction without the need for manual delineation.

### 2.2.1 Comparison to manual delineation

The accuracy of the data product is estimated by comparing automated calving front extractions to manual delineations. Loebel et al. (2024c) have already evaluated the processing system for accuracy and generalizability, with particular emphasis on Greenland Glaciers. Since we use additional training data for this analysis, we also apply a manually delineated test dataset specifically for the AP. This test dataset contains 60 calving front locations over 20 glaciers. This includes additional eight glaciers which are not part of the training dataset. These additional eight test glaciers ensure the spatial transferability of our method. Whereas the training data contains calving fronts from 2013 to 2021, the test dataset contains calving fronts for the separate period from 2022 and 2023. As ANN training is not deterministic, we train five separate models for our assessment. Our main error metric is the distance between the predicted delineation and the manual delineation. For this we implement two different distance estimates. Firstly, we use an average minimal distance error which we calculate by averaging the minimum

**Table 2.** Results of the accuracy assessment presented as mean values with corresponding standard deviations calculated over the five trained models. The average minimal distance and the Hausdorff distance estimates are provided as mean and median values over the test dataset. Additionally shown are accuracy, precision, recall and F1-Score are given. Binary classification metrics relate to the land/glacier class.

| Average minimal distance | | Hausdorff distance | | Binary classification metrics | | | |
|---|---|---|---|---|---|---|---|
| Mean (m) | Median (m) | Mean (m) | Median (m) | Accuracy | Precision | Recall | F1-Score |
| $59.3 \pm 5.9$ | $33.9 \pm 1.5$ | $405.1 \pm 20.7$ | $257.0 \pm 14.7$ | $0.984 \pm 0.001$ | $0.978 \pm 0.002$ | $0.995 \pm 0.001$ | $0.986 \pm 0.001$ |

distance every $30\,\mathrm{m}$ along the predicted front trajectory. This estimate is comparable to the ones used by Cheng et al. (2021), Loebel et al. (2022), Baumhoer et al. (2023) and Zhang et al. (2023). Secondly, we report the Hausdorff distance (Huttenlocher et al., 1993) which only considers the largest distance of all minimal distances along the two trajectories. The Hausdorff distance is therefore very sensitive to discrepancies even along small sections of the glacier front between the ANN and the manually derived calving front.

Figure 3 shows six test images for a diverse range of challenging conditions concerning ice mélange, cloud cover, iceberg presence, low illumination and satellite scene borders. In addition, the minimal distance along the predicted trajectory (from A to B) is shown for each image. Our processing system reliably delineates calving fronts from a wide range of image conditions. These include a wide range of ocean, ice mélange and illumination conditions, light cloud cover, and images with calving fronts near the edge of a satellite scene. This is due to the large training dataset, which covers a wide variety of satellite images under these conditions. In addition, the integration of multispectral input data leads to more accurate predictions under these difficult situations than using only single bands inputs (Loebel et al., 2022). Looking at the distance error along the predicted fronts, it is clear that the difference between manual delineation and ANN is not uniform. This is also reflected in the Hausdorff and average minimum distance errors, which can vary widely. Large minimal distance errors mostly occur due to inaccurate ANN predictions at difficult-to-delineate parts of the glacier front (e. g. at ~11 km in Fig. 3 (f) or at ~15 km in Fig. 3 (c)). However, it is important to note, that although we treat it as such here, our manual delineation is not a traditional ground truth, as it is also uncertain depending on the author and the satellite image. For difficult-to-delineate scenes (e. g. Prospect Glacier in Fig. 3 (d)), it is not possible to attribute the error as both the manual and ANN predictions are uncertain.

Table 2 gives an overview of the accuracy assessment over the entire test dataset. In addition to the average minimal distance and the Hausforff distance estimates, we also specify the binary classification metrics accuracy, precision, recall and F1-Score. Whilst a high binary classification performance does not necessarily translate to an accurate prediction of the calving front trajectory, we report these values to facilitate comparability of our results with other studies and datasets. Although completely different test datasets are involved, the $59.3 \pm 5.9\,\mathrm{m}$ mean average minimal distance calculated here aligns very well with the $61.2 \pm 7.5\,\mathrm{m}$ reported by Loebel et al. (2024c). When applied to the ESA-CCI (ENVEO, 2017) and CALFIN (Cheng et al., 2021) test datasets (as processed in Loebel et al. (2024c)), which contain a further 100 and 110 additional test images from Greenland glaciers, respectively, we calculate a mean average minimal distance of $79.1 \pm 5.3\,\mathrm{m}$ and $78.7 \pm 3.8\,\mathrm{m}$ (see extended accuracy assessment table in supplement). Furthermore these results are in the broad range of other ANN-based calving front

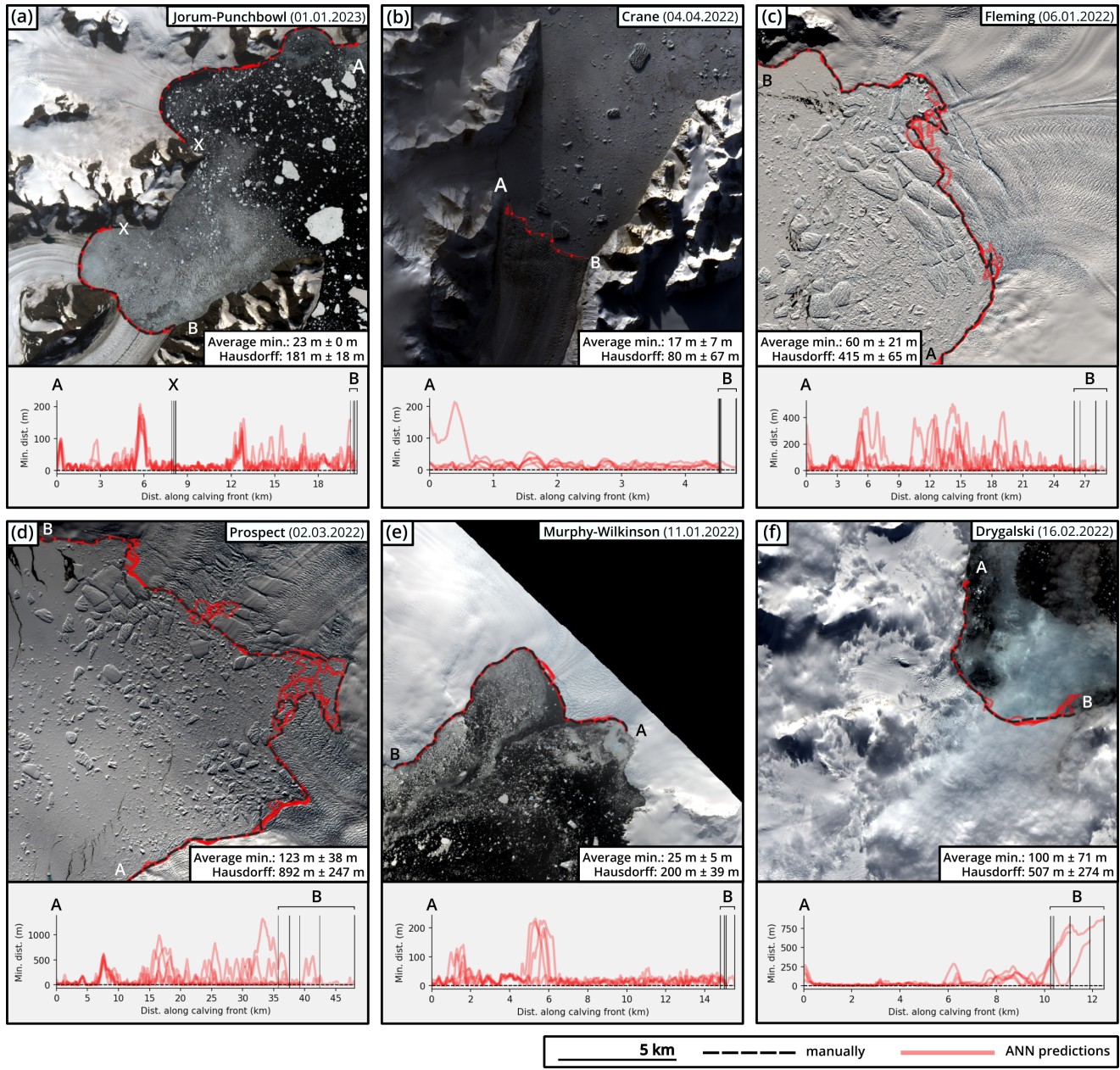

**Figure 3.** Accuracy assessment results of 6 sample scenes from the test dataset. Dashed black lines show manually delineated calving fronts. Graphs at the bottom of each image show the minimal distance along the predicted trajectory (from A to B). Red lines show the five ANN predictions from five models. Overlap of lines is indicated by higher color intensity. Average minimal distance and Hausdorff distances are given for each test image. Note that the endpoints B of different models do not coincide due to the different lengths of the predicted fronts. For location of specific glaciers, see Figure 1. Landsat imagery courtesy of the U.S. Geological Survey.

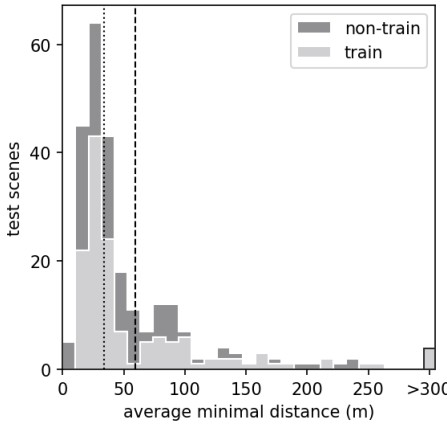

**Figure 4.** Histogram of the average minimal distance between manual delineation and ANN prediction. The number of total test scenes corresponds to the test dataset multiplied by the 5 trained ANN models. The overall median is shown as a dotted line and the overall mean as a dashed line. Grey levels indicate whether the test scene is from a glacier that is part of the training dataset or not.

extraction methods using optical imagery like Cheng et al. (2021) with $86.7 \pm 1.4\,\mathrm{m}$ and Zhang et al. (2023) with $79\,\mathrm{m}$. The Hausdorff distance is commonly not reported in other automated glacier front delineation studies. However, Goliber et al. (2022) estimated the error in manual delineation by applying the median Hausdorff distance to duplicate delineations from different authors. Depending of the pairs of authors median errors range from $58.6\,\mathrm{m}$ to $7350\,\mathrm{m}$ with an overall median error of $107\,\mathrm{m}$. This suggests that our method, which has a median Hausdorff error of $257 \pm 14.7\,\mathrm{m}$, is in the range of possible manual

delineation errors, but has not yet reached human performance.

When assessing the accuracy only for the 23 test scenes of glaciers outside the training dataset we calculate a mean delineation error of $51.9 \pm 6.7\,\mathrm{m}$ (median: $37.3 \pm 5.3\,\mathrm{m}$). Interestingly, the mean is lower and the median is higher than for an assessment over the 37 scenes from glaciers within the training dataset, where we mean (and median) delineation error is $65.3 \pm 7.7\,\mathrm{m}$ (median: $33.8 \pm 1.5\,\mathrm{m}$). This is likely because of training glaciers that have challenging-to-delineate calving con-

175 ditions (like Prospect Glacier, see Fig. 3 (d)). Figure 4 shows the distribution of the average minimal distance error accumulated over all test scenes for the 5 trained models. Based on these numbers, we confirm the high degree of ANN model generalization and hence the spatial transferability of our method.

### 2.2.2 Inter-model distance

As a secondary estimate of accuracy, we introduce the inter-model distance. We define the inter-model distance as the average

minimal distance between two predicted calving front trajectories from two different ANN models that have the same architecture and use the same set of hyperparameters. Specifically for our case, we calculate a mean inter-model distance between the ANN model we use to generate our data product (randomly selected) and the four other trained models. The concept behind this metric is that the variability of different models predicting the same calving front is highly correlated with the accuracy of

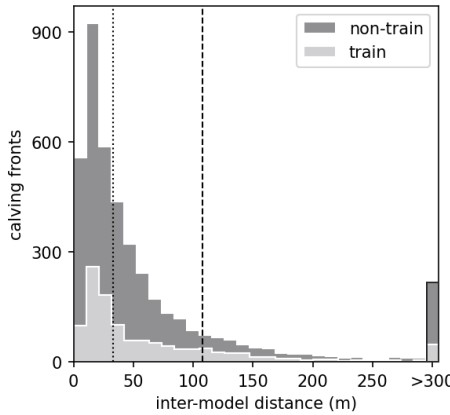

**Figure 5.** Histogram of the mean inter-model distance over all calving front entries in our data product. The overall median is shown as a dotted line and the overall mean as a dashed line. Grey levels indicate whether the test scene is from a glacier that is part of the training dataset or not.

those predictions. The main motivation for introducing the mean inter-model distance is that it can be calculated for each calving front prediction without the need for manual delineation. Hence, the mean inter-model distance can be reported for each entry in our data product. However, limitations need to be considered when evaluating this metric. Although a low inter-model distance confirms that the ANN is confident in its prediction, it does not necessarily translate into an accurate calving front. Systematically incorrect predictions from all five ANN models (e.g. due to an iceberg being misclassified as a glacier) result in a low mean inter-model distance, when in fact the predicted front would be of low quality with a large distance to manual delineation. That said, as incorrect predictions are mostly eliminated during quality control of the data product (see Sect. 2), the opposite is more likely to happen. Here, a high mean inter-model distance is calculated, although the predicted front is of good quality. This happens most often in satellite scenes with difficult image conditions where the ANN predictions have low confidence, causing some models to delineate the front correctly. An accurately delineated calving front prediction can therefore yield a high mean inter-model distance when at least one of the predictions of the other four models are of low quality. Overall, we believe that this mean inter-model distance is a useful estimate of accuracy, although it will likely overestimate the delineation error for some calving front predictions.

Figure 5 shows the distribution of the mean inter-model distance over all calving front entries in our data product. Again, we distinguish whether the entire is from a glacier that is part of the training dataset or not, further validating spatial transferability. Across our entire data product we calculate a mean of $107.9\,\text{m}$ and a median of $33.2\,\text{m}$. While the mean value is significantly higher than the mean average minimal distance of our test dataset, the median values are almost the same. It should also be noted that the mean inter-model distance varies considerably from glacier to glacier. Separate histograms for each of the 42 glaciers are included in the supplement.

**Table 3.** Temporal coverage of our ANN generated time series. The numbers and the color intensity indicate the amount of processed calving front positions in the respective year.

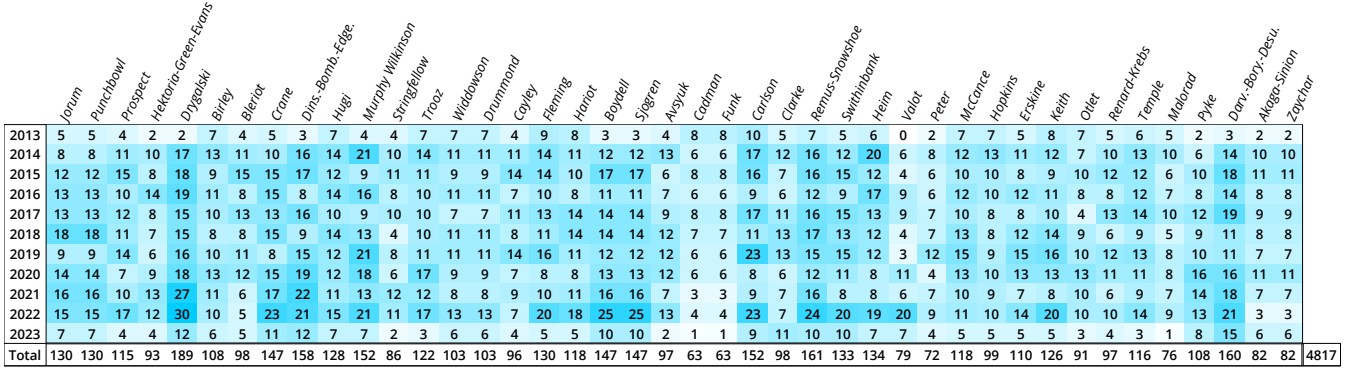

| Year | Jorum | Punchbowl | Prospect | Hektoria-Green-Evans | Drygalski | Birley | Bleriot | Crane | Dins.-Bomb.-Edge. | Hugi | Murphy Wilkinson | Stringfellow | Trooz | Widdowson | Drummond | Cayley | Fleming | Hariot | Boydell | Sjögren | Airsyuk | Cadman | Funk | Carlson | Clarke | Remus-Snowshoe | Swithinbank | Heim | Valot | Peter | McCance | Hopkins | Erskine | Keith | Ollet | Renard-Krebs | Temple | Malorad | Pyke | Drav.-Bory-Desu. | Akoga-Sinion | Zeychar | Total |
|---|---|---|---|---|---|---|---|---|---|---|---|---|---|---|---|---|---|---|---|---|---|---|---|---|---|---|---|---|---|---|---|---|---|---|---|---|---|---|---|---|---|---|---|
| 2013 | 5 | 5 | 4 | 2 | 2 | 7 | 4 | 5 | 3 | 7 | 4 | 4 | 7 | 7 | 7 | 4 | 9 | 8 | 3 | 3 | 4 | 8 | 8 | 10 | 5 | 7 | 5 | 6 | 0 | 2 | 7 | 7 | 5 | 8 | 7 | 5 | 6 | 5 | 2 | 3 | 2 | 2 | |
| 2014 | 8 | 8 | 11 | 10 | 17 | 13 | 11 | 10 | 16 | 14 | 21 | 10 | 14 | 11 | 11 | 11 | 14 | 11 | 12 | 12 | 13 | 6 | 6 | 17 | 12 | 16 | 12 | 20 | 6 | 8 | 12 | 13 | 11 | 12 | 7 | 10 | 13 | 10 | 6 | 14 | 10 | 10 | |
| 2015 | 12 | 12 | 15 | 8 | 18 | 9 | 15 | 15 | 17 | 12 | 9 | 11 | 11 | 9 | 9 | 14 | 14 | 10 | 17 | 17 | 6 | 8 | 8 | 16 | 7 | 16 | 15 | 12 | 4 | 6 | 10 | 10 | 8 | 9 | 10 | 12 | 12 | 6 | 10 | 18 | 11 | 11 | |
| 2016 | 13 | 13 | 10 | 14 | 19 | 11 | 8 | 15 | 8 | 14 | 16 | 8 | 10 | 11 | 11 | 7 | 10 | 8 | 11 | 11 | 7 | 6 | 6 | 9 | 6 | 12 | 9 | 17 | 9 | 6 | 12 | 10 | 12 | 11 | 8 | 8 | 12 | 7 | 8 | 14 | 8 | 8 | |
| 2017 | 13 | 13 | 12 | 8 | 15 | 10 | 13 | 13 | 16 | 10 | 9 | 10 | 7 | 7 | 11 | 13 | 14 | 14 | 14 | 9 | 8 | 8 | 17 | 11 | 16 | 15 | 13 | 9 | 7 | 10 | 8 | 8 | 10 | 4 | 13 | 14 | 10 | 12 | 19 | 9 | 9 | 9 | |
| 2018 | 18 | 18 | 11 | 7 | 15 | 8 | 8 | 15 | 9 | 14 | 13 | 4 | 10 | 11 | 11 | 8 | 11 | 14 | 14 | 14 | 12 | 7 | 7 | 11 | 13 | 17 | 13 | 12 | 4 | 7 | 13 | 8 | 12 | 14 | 9 | 6 | 9 | 5 | 9 | 11 | 8 | 8 | |
| 2019 | 9 | 9 | 14 | 6 | 16 | 10 | 11 | 8 | 15 | 12 | 21 | 8 | 11 | 11 | 11 | 14 | 16 | 11 | 12 | 12 | 12 | 6 | 23 | 13 | 15 | 15 | 12 | 3 | 12 | 15 | 9 | 15 | 16 | 10 | 12 | 13 | 8 | 10 | 11 | 7 | 7 | 7 | |
| 2020 | 14 | 14 | 7 | 9 | 18 | 13 | 12 | 15 | 19 | 12 | 18 | 6 | 17 | 9 | 9 | 7 | 8 | 8 | 13 | 13 | 12 | 6 | 6 | 8 | 6 | 12 | 11 | 8 | 11 | 4 | 13 | 10 | 13 | 13 | 13 | 11 | 11 | 8 | 16 | 16 | 11 | 11 | |
| 2021 | 16 | 16 | 10 | 13 | 27 | 11 | 6 | 17 | 22 | 11 | 13 | 12 | 12 | 8 | 8 | 9 | 10 | 11 | 16 | 16 | 7 | 3 | 3 | 9 | 7 | 16 | 8 | 6 | 7 | 10 | 9 | 7 | 8 | 10 | 6 | 9 | 7 | 14 | 18 | 7 | 7 | 7 | |
| 2022 | 15 | 15 | 17 | 12 | 30 | 10 | 5 | 23 | 21 | 15 | 21 | 11 | 17 | 13 | 13 | 7 | 20 | 18 | 25 | 25 | 13 | 4 | 4 | 23 | 7 | 24 | 20 | 19 | 20 | 9 | 11 | 10 | 14 | 20 | 10 | 10 | 14 | 9 | 13 | 21 | 3 | 3 | |
| 2023 | 7 | 7 | 4 | 4 | 12 | 6 | 5 | 11 | 12 | 7 | 7 | 2 | 3 | 6 | 6 | 4 | 5 | 5 | 10 | 10 | 2 | 1 | 1 | 9 | 11 | 10 | 10 | 7 | 7 | 4 | 5 | 5 | 5 | 5 | 3 | 4 | 3 | 1 | 8 | 15 | 6 | 6 | |
| Total | 130 | 130 | 115 | 93 | 189 | 108 | 98 | 147 | 158 | 128 | 152 | 86 | 122 | 103 | 103 | 96 | 130 | 118 | 147 | 147 | 97 | 63 | 63 | 152 | 98 | 161 | 133 | 134 | 79 | 72 | 118 | 99 | 110 | 126 | 91 | 97 | 116 | 76 | 108 | 160 | 82 | 82 | 4817 |

## 3 Data product and usage notes

The data product presented here has been created to provide glaciologists and glacier modellers with high quality calving front positions of the AP Ice Sheet without the need for manual delineation. Figure 1 gives a spatial overview of the 42 processed glaciers. A tabular overview is given in Table 3. In total the data record encompasses 4817 calving front positions over 42 marine-terminating glaciers. Since the data is derived from optical imagery, the time series have a 14-week gap during polar night from May to mid-August. Outside polar night, the dataset has one entry every 19.5 days on average. However, the sampling is irregular and primarily dependent on the satellite orbit and cloud cover. The time frame from 2013 to 2023 covers that of the IceLines dataset (Baumhoer et al., 2023), facilitating a combined analysis of circum-Antarctic calving front change.

Figure 6 gives eight example time series of terminus area change within two regions of the AP. The terminus area change of glaciers in the Larsen-B embayment (Fig. 6 a-d) is spatially correlated and shows a steady advance from 2013 until the end of 2021. At the beginning of 2022, our data show a simultaneous retreat of the four glaciers. Subsequently, the glacier tongues of Hektoria-Green-Evans, Jorum, and Crane glacier have collapsed. This simultaneous retreat is attributed to the disintegration of landfast sea ice inside the embayment in early 2022 and the resulting loss of buttressing (Ochwat et al., 2024). The glaciers in Wordie Bay (Fig. 6 e-h) show a more varied calving front dynamics. These range from stable calving front positions (Hariot Glacier and Carlson since mid 2020) over steady terminus advance superimposed by frequent calving events (Fleming Glacier) to large calving events (Prospect Glacier in 2018). The dynamic changes in this area are linked to the Wordie Ice Shelf and its disintegration between the 1960s and the late 1990s. This has led to an increased ice flow and calving of the three main tributary glaciers Hariot, Fleming, Carlson and Prospect (Friedl et al., 2018). Therefore, an operational and temporally high resolution monitoring of these glaciers is particularly important. An overview of the the time series of all 42 glaciers of our data product is given in the supplement.

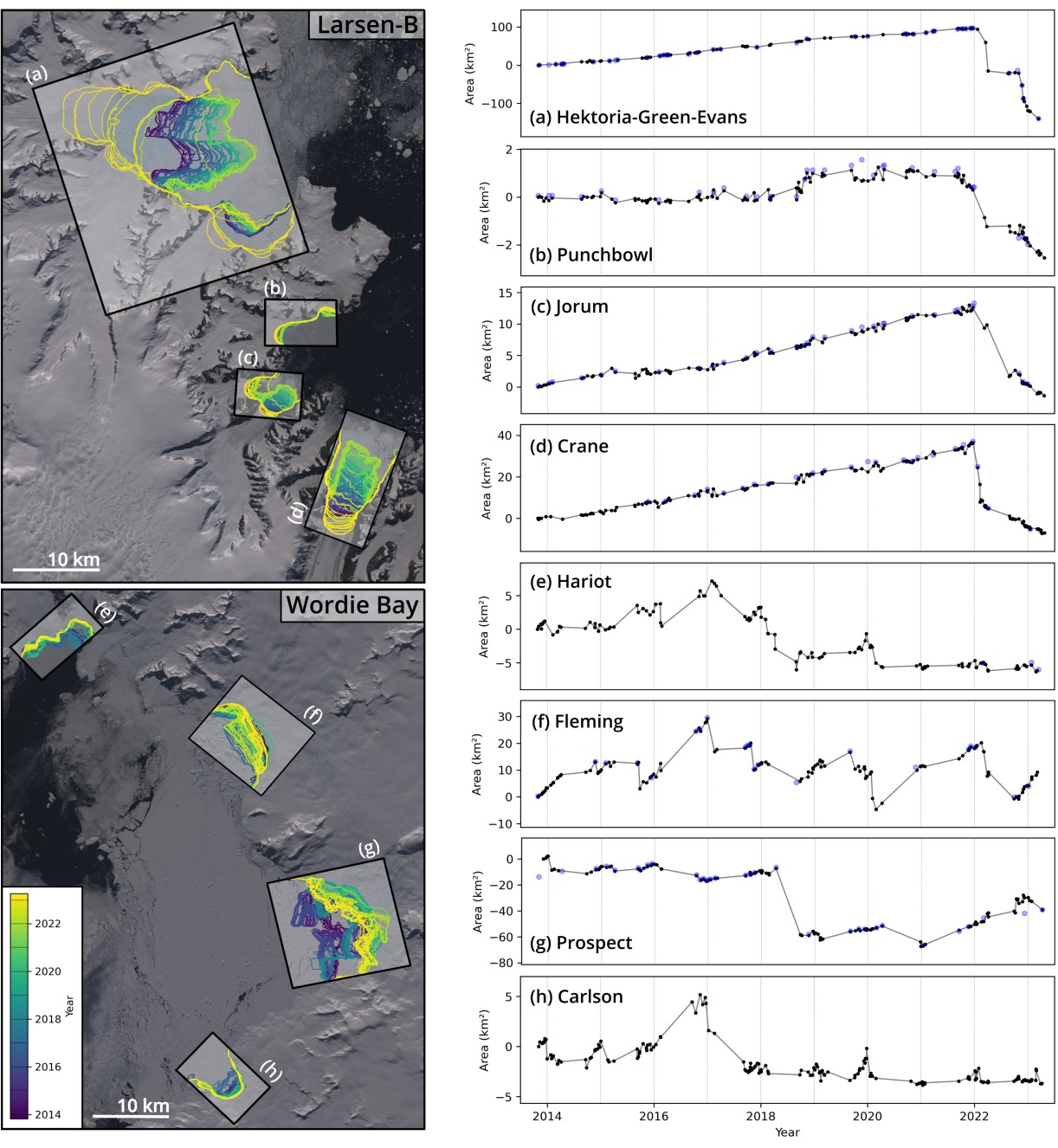

**Figure 6.** Example time series of terminus area change generated by our processing system for (a-d) four AP glacier in the Larsen-B embayment and (e-g) three glaciers at Wordie Bay. Color-coded calving front locations are depicted in the maps in the left. Corresponding time series are shown on the right with entries marked by black dots. The blue dots are additional validation marks that indicate the frontal positions of the manually delimited reference dataset. Landsat imagery courtesy of the U.S. Geological Survey.

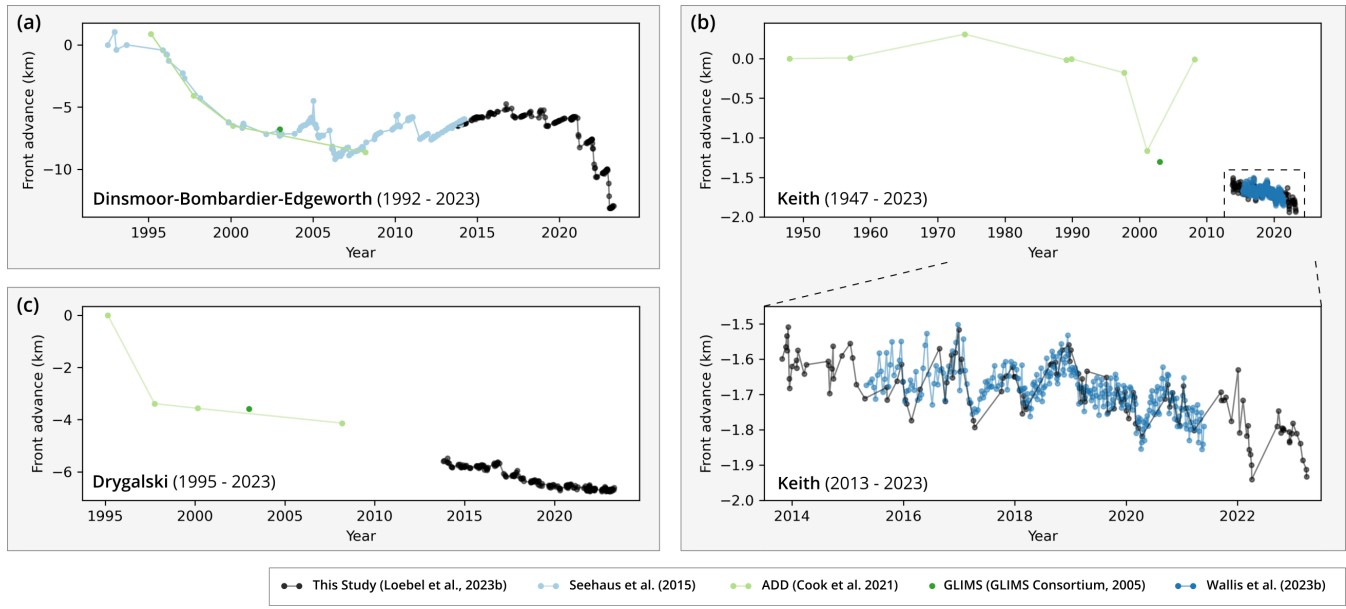

**Figure 7.** Time series of glacier advance along central flowline for (a) the Dinsmore-Bombardier-Edgeworth glacier system, (b) Keith Glacier and (c) Drygalski Glacier. The individual datasets are marked with different colours. Note that the axes have differing scales.

To put our data product in context with existing datasets, we contrast the different time series. Figure 7 shows time series of glacier advance for three examples. For each glacier, all available datasets were used (see Tab. 1). For the Dinsmoor-
Bombardier-Edgeworth glacier system (Fig. 7 (a)), there is generally very good agreement between the four datasets. Here, our data product provides a valuable continuation of the time series of Seehaus et al. (2015), which was delineated as part of a glaciological study (Seehaus et al., 2015) for this glacier. Similarly, for the Keith Glacier (Fig. 7 (b)), the dataset of Wallis et al. (2023a) has significant overlap with our time series. Although calving front change during this period is relatively small, the seasonal and subseasonal variations are captured by both datasets. Importantly differences between these time series are in the
range of both manual and ANN delineation accuracy. The time series of Drygalski Glacier (Fig. 7 (c)) is representative for the majority of glaciers in our data product. Here our time series is the first seasonal record, with the other available datasets being the GLIMS database (GLIMS Consortium, 2005) and the ADD (Cook et al., 2021b). These three examples highlight that the quality of our automatically delineated calving fronts is comparable to existing manually extracted datasets. As a result, our data product is the first to combine seasonal temporal resolution with a large spatial coverage across the AP. Nevertheless, the
importance of the GLIMS and ADD products must be emphasised, as these are still the only repositories that provide complete coverage and, the ADD in particular, long term observations ranging back to 1843.

The data product is stored in a georeferenced vector file format (GeoPackage and ESRI Shapefile), sorted by glacier and date within a file system structure. All files are georeferenced using the Antarctic Polar Stereographic Projection (EPSG:3031). This allows for easy handling, e.g. by means of GIS software or geospatial data libraries. Calving front traces are stored separately
for each glacier and each date, as well as in a consolidated file. In addition to the full data product, annual entries are provided

for each glacier. Annual entries also contain the full ANN coastline predictions which are provided as both linestings and polygonal masks. The attribute table of each file includes the glacier name, calving front date, type (glacier front or coastline), processing date, processing version, corresponding Landsat product identifier, mean inter-model distance and standard deviation of the inter-model distance. The file naming convention for each entry is: *[glacier name]_[YYYYMMDD]_[type].shp*. An example entry would be: *prospect_20230408_glacier_front.shp*.

## 4 Data and code availability

The AP calving front location data record is publicly available at PANGAEA under https://doi.pangaea.de/10.1594/PANGAEA.963725 (Loebel et al., 2023). The calving front locations can be downloaded by clicking on the "View Dataset as HTML" button in the overview. All reference data applied in this study is publicly available. Greenland reference data is available at http://dx.doi.org/10.25532/OPARA-282 (Loebel et al., 2024d) and AP reference data is available at https://doi.org/10.25532/OPARA-581 (Loebel et al., 2024b). We provide a containerized implementation (platform: Docker) of the presented processing system. The software automatically extracts calving front positions from Landsat-8 or Landsat-9 Level-1 data archives for glaciers used within this study or at user-defined coordinates. This enables the analysis of glaciers that are outside our reference dataset or beyond the temporal frame of our study. The software is available at https://github.com/eloebel/glacier-front-extraction (last access 24 July 2024) and https://doi.org/10.5281/zenodo.7755774 (Loebel, 2023a). Our implementation (software: Python 3) of the rectilinear box method is available at https://github.com/eloebel/rectilinear-box-method (last access 24 July 2024) and https://doi.org/10.5281/zenodo.7738605 (Loebel, 2023b). The processed time series of terminus area change, provided in text file format, are available at https://doi.org/10.25532/OPARA-557 (Loebel et al., 2024a).

## 5 Conclusions

Accurate as well as temporally and spatially comprehensive calving front data is essential for understanding and modelling glacial evolution. This paper addresses this requirement and presents a new data record for glaciers at the AP. The data is generated by applying multispectral Landsat-8 and Landsat-9 imagery to a deep learning based processing system. We validated the processing system for accuracy, robustness, and generalization capabilities using independent test data. The mean difference between automated and manual extraction is estimated at $59.3 \pm 5.9\,\mathrm{m}$. The resulting data record contains 4817 calving front locations for 42 key outlet glaciers from 2013 to 2023. It achieves sub-seasonal temporal resolution for all processed glaciers, making it a valuable addition to existing data records.

More broadly, this contribution exemplifies that well generalised ANN processing systems can be applied to various regions of interest with only minor additions to reference data. With thousands of marine-terminating glaciers worldwide (RGI Consortium, 2017), this is particularly relevant for extracting calving fronts. In addition to applications for current satellite missions, there is also significant potential for improving historical data records by exploiting the vast amount of satellite imagery col-

lected over the past decades. We expect that our presented data record will not only advance glaciological research for the AP, but also contribute to future deep learning based calving front data products and data inter-comparison projects.

*Author contributions.* Contributions are according to CRediT. Conceptualization: EL, CAB, AD and MS. Data curation: EL. Formal analysis: EL. Funding acquisition: MH, MS, AD. Investigation: EL. Methodology: EL. Software: EL. Supervision: CAB, AD, MH, MS. Validation: EL. Visualization: EL. Writing (original draft): EL. Writing (review and editing): EL, CAB, MH and MS.

*Competing interests.* The author declares that there are no competing interests.

*Acknowledgements.* We thank the USGS for providing Landsat-8 and Landsat-9 imagery. We are grateful to the TU Dresden Center for Information Services and High Performance Computing (ZIH) for providing their high-performance and storage infrastructure. We acknowledge the Norwegian Polar Institute's Quantarctica package. This work was supported by the Helmholtz Association of German Research Centers as part of the Helmholtz Information and Data Science Incubator, project "Artificial Intelligence for Cold Regions" (AI-CORE, grant no. ZT-I-0016), and by the German Federal Ministry of Education and Research (BMBF), project "Greenland Ice Sheet/Ocean Interaction" (GROCE2, grant no. 03F0778G).

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
