# Peer review of "Calving front positions for 42 key glaciers of the Antarctic Peninsula Ice Sheet: a sub-seasonal record from 2013 to 2023 based on a deep learning application to Landsat multispectral imagery"

_Earth System Science Data, 2023_

## Referee Comment (RC1)

**Review of Loebel et al. (2024): Calving front positions for 19 key glaciers of the Antarctic Peninsula: a sub-seasonal record from 2013 to 2023 based on a deep learning application to Landsat multispectral imagery**

**General comments:**

Firstly, I am not an expert in machine or deep learning techniques, and I can see that the underlying method has already been described in detail in Loebel et al. (2022) and applied to 23 Greenland glaciers in Loebel et al. (2023, in review). I will therefore focus my review on the (1) the dataset itself and the aspects of the methodology relevant to producing a time-series of calving front positions for the community and, (2) the accuracy assessment of the method.

The paper presents an exciting application of an existing deep learning method for delineating glacier calving fronts to 19 glaciers on the Antarctic Peninsula. Compared to Greenland, relatively few terminus position datasets are available for the Antarctic Peninsula and the generation of new terminus positions has not kept pace with the generation of new velocity measurements. As a potential future user of this dataset, it is great to see this application and I am confident that new terminus position delineations on the Peninsula will benefit the community. As such, I am wholeheartedly in support of the generation and publication of these datasets. However, I do not yet think that the presented dataset or manuscript meets the quality and scope required for publication in ESSD, but I am hopeful that the authors will take on board my criticisms and suggestions so that this manuscript and dataset can meet the needs of the community and make best use of the deep learning tool that the author has developed.

**Specific comments:**

1a) The scope of the dataset

The authors present a total 2064 calving front delineations across 19 outlet glaciers from 2013 to 2023. One the big questions I had after reading the manuscript was "why not more?". Just to be clear, I don't wish to belittle the efforts of the authors – I am sure it is a lot of work to do this and I know it is a lot of work to generate new datasets. However, there are 1,728 basins in the Cook et al. (2014) basin dataset, roughly half of which terminate in an ice shelf, so there are perhaps 800-odd glaciers on the Peninsula that could be targeted by this method. Since the deep learning method was already developed and the majority of the training dataset already existed, and because comparisons to regions outside of Greenland have already been presented in Loebel et al. (2023), it seems like a relatively small additional contribution to run the processing system for just 19 glaciers, especially given that ESSD does not demand any analysis seeking to develop new understanding from the presented dataset, which is typically the bulk of the work in other journals. Again, I am sure it was a lot of work to do this, which I don't want to detract from, but one of the key benefits of the method used in the manuscript is that it is automatic and much faster than manual approaches, so it should be able to provide "additional and more comprehensive data products". Therefore, I don't think it is sufficient to present a terminus position dataset that is (for example) ~25% smaller than that in Wallis et al. (2023), given that the dataset in Wallis et al. (2023) was a relatively small component of their publication. It would be great to see a definitive dataset of terminus positions for the Antarctic Peninsula over the last decade – this and the lead author's earlier papers demonstrate that we now have the tools and imagery available to achieve this, so I think that is something we should strive for. In order for this dataset to be suitable for publication in ESSD, and to really demonstrate the utility of the underlying deep learning method, I strongly suggest that it should be applied to many more glaciers on the Antarctic Peninsula.

If there is a good scientific or resource reason for limiting the analysis to a small subset of glaciers, then I would still argue for a larger subset including other major glaciers (e.g. Cadman Glacier, which seems like a major omission here), and I think that more justification for the choice of glaciers should be given. At present, the choice is justified twice in the paper, but only briefly and different reasons are given each time.

**1b) Filtering of 'raw' terminus positions**

One of the main focuses of this paper is that it generates new time-series of terminus positions from an existing method. I was surprised therefore that the manuscript didn't describe much post-processing of the terminus positions in order to make an analysis-ready time-series. The only filtering step I could see is that the authors "separate all entries that have an area difference of more than 1 km$^2$ from the previous and following entries". I don't think that is a sufficiently robust outlier removal technique, especially if you choose to apply this to more glaciers. I suggest that the outlier removal routine should (1) account for the speed of the glacier and time separation between measurements; (2) account for the width of the glacier, because 1 km$^2$ changes might be realistic or not depending on their width, and; (3) account for changes along flowlines or similar, not just width-averaged metrics. The time-series presented in the manuscript look reasonably clean, but I had a quick look at the dataset which showed up some places where I think the outlier identification may not be working, for example Birley glacier on 2022/10/03 has what looks like an unrealistic advance along it's southern branch and Sjogren-Boydell on 2022/03/04 has a ~4 km retreat across what I think is a large section of land. Whichever outlier removal approach is used, the authors should provide details of how many delineations are removed through using it and how that affects the results.

On a related note, I couldn't see any description in the manuscript of partial terminus positions. Does the deep learning approach always provide a full terminus trace? What if the glacier terminus is partially obscured by clouds? Please add detail to the manuscript as necessary.

**1c) Vectorization of the land/ice probability masks**

I couldn't see much justification for the choice of a 0.5 threshold or the impact of that choice of the resulting terminus location. This might be described in the author's earlier papers, but I think ESSD is a suitable place to provide more detail and I think it is relevant to the terminus dataset. Firstly, I think the chosen threshold should be clearly stated in the text, rather than only in the figure (apologies if I just missed it). Secondly, I think there should be a clear quantification, and ideally visualisation, of the impact of that threshold on individual terminus locations and the resulting time-series of area change.

**1d) Error metric for predicted delineations**

Could you provide an error metric for each individual delineation, perhaps by using the spread amongst the 5 models and/or the spread amongst different thresholds? I'm aware that such errors are not provided for manual delineations because it is impractical, but it seems achievable and useful for this method, especially given the apparent differences between each of the 5 models on challenging images shown in Figure 3.

**1e) Dataset format**

I think that dataset would be much easier and quicker to use if there was just one shapefile for each glacier plus one shapefile containing all delineations for all glaciers. Some users are now also using the geopackage format, so the authors might want to consider providing the output in that format also.

**1f) Dataset contents**

It would be great if the training and test data were also released, along with the automatic delineations.

Some glaciers and times seem to be missing a 'coastline' shapefile. Is that expected? Also, the justification in the manuscript for providing two outputs is not clear. Why is the coastline file better than the glacier file for merging with an ice mask? It's implied that the terminus file contains only the glacier edge, whereas the coastline file contains the glacier edge and the edge of the surrounding fjord walls, but this isn't demonstrated clearly nor how that distinction is made if part of the terminus is

obscured by clouds, for example. Please add some clarification and justification in that respect to the manuscript.

2a) Accuracy assessment

If I have understood it corrected, the accuracy assessment consists of a comparison between the 5-model mean delineation (with a single threshold) and three manual delineations per glacier outside of the training window. Only one of those images for 10 of the 19 glaciers is shown and otherwise we are provided with some simple metrics summarising the results of 19x3 comparisons. In my view, that is not sufficient to characterise the accuracy of the model in this region. Figure 3 is a useful illustration (though I have some suggestions below), but there is no evidence given that demonstrates that the examples given in Figure 3 are representative of the typical accuracy of the method under those conditions, or what the spread in performance is like in each of those conditions. I suggest that the accuracy assessment should include enough images to provide statistically significant accuracy measures of accuracy for glaciers and images with each of the different characteristics shown in Figure 3. Ideally, it would also show the effect of combinations of those conditions, such as times of low illumination with a scene border, with or without mélange and cloud cover.

Figure 3: this is one of the main pieces of evidence presented in the manuscript to convince the reader that the automatic delineations performs as well as a manual delineation. However, showing ~15x15 km images to illustrate differences in position of less than 100 m is not a very clear way to illustrate those differences – the figures would need to be produced at an impractical resolution and I would need a much better monitor to see anything meaningful, and even then I wouldn't be able to measure the differences. I suggest that an additional, more quantitative figure should be provided, to show differences between the automatic and manual delineations for the full test dataset. Perhaps some simple graphs with 'distance along terminus' on the x-axis and 'difference from manual delineation' on the y-axis would allow the authors to plot the differences through the full test dataset every 30 m along each delineation? That kind of plot would also clearly show how those differences are affected by your choice of model and threshold for vectorization. You could have one graph per glacier and perhaps show a histogram for each glacier.

I am unsure that the accuracy metrics of mean and median difference from manual delineations is representative of the differences between the automatic and manual delineations with regard to evaluating the use of the automatic delineations for scientific purposes. As far as I can tell, both of those metrics would be insensitive to large differences between automatic and manual delineations if those differences occur over a short section of the terminus. For example, Figure 3 shows automatic delineations on Prospect are in several places over 1 km from the manual delineation, but the mean difference is small because there are comparatively long sections where the two sets of delineations are in close agreement. This shows up a bit in the Prospect timeseries in Figure 4g, where there is a ~10 km$^2$ difference between the automatic and manual delineation in late-2022. For glaciological and modelling applications, it might be that those areas of large difference are the bits that matter, so the mean error across the terminus wouldn't be a useful error metric. The other problem with this is that the mean or median difference between the delineations will be highly dependent the length of glacier terminus compared to the length of non-glacier digitised coastline.

As presented, there isn't a compelling demonstration that the dataset is as applicable to science cases than manually-derived datasets, which I think is important given the proposed justification for making the dataset. Another more holistic approach the authors should take to demonstrate the quality of their time-series product, which would go a long way to addressing that concern, would be to compare time-series of area change from these new delineations to area change time-series derived from other terminus position datasets, where both/multiple datasets have sampled the same glacier during overlapping time periods.

3) Introduction

I do not think the introduction adequately justifies the need for improved monitoring of outlet glacier terminus position change. As written, it states that (1) ice shelves have reduced in thickness and extent, which has led to glacier speed-up, (2) calving fronts can be used to study ice-ocean interaction and that (3) they can be used to improve model simulations. Those points are all true, but I think they need more detail and specifics in order to make a convincing argument for this new dataset. Consider including more detail of ice shelf and glacier area changes (citing the various papers by Cook et al on the subject) and how much the Peninsula has contributed to Antarctica's total sea level contribution. Are there specific examples of where model performance has been limited or improved by the availability of calving front positions, or has it been quantified in a more general sense? For such models, I think they would they need a continual coastline across the whole domain, not small subsets as provided here. In addition, consider drawing on the literature from Greenland, where measurements of terminus position change have led improvements in our understanding of glacier response to environmental conditions over a range of spatial and temporal scales (e.g. Cowton et al., 2018) or have at least aided the interpretation of changes in ice speed, and how they have been used in combination with estimates of submarine melt rates to develop new parameterisations for the impact of submarine melting on calving and terminus position (Slater et al., 2019). Terminus positions are really useful, but I don't think that comes across in the introduction as currently written.

Cowton, T.R., Sole, A.J., Nienow, P.W., Slater, D.A. and Christoffersen, P., 2018. Linear response of east Greenland's tidewater glaciers to ocean/atmosphere warming. Proceedings of the National Academy of Sciences, 115(31), pp.7907-7912.

Slater, D. A., Straneo, F., Felikson, D., Little, C. M., Goelzer, H., Fettweis, X., and Holte, J.: Estimating Greenland tidewater glacier retreat driven by submarine melting, The Cryosphere, 13, 2489–2509, https://doi.org/10.5194/tc-13-2489-2019, 2019.

**Minor comments**

Line 4: suggest "rely on manual delineation, **which is** time-consuming"

Line 8: suggest "The data product presented here"

Line 16: ice **shelf**

Line 17: "forcing from ocean and atmosphere has led to reduced ice shelf thickness and extent. And this, in turn, has reduced buttressing strength and thereby increased outlet glacier dynamics". I don't think this is a fair summary of our current understanding of ice shelf and glacier changes on the Peninsula. Can you add more detail on what is meant by "forcing". I don't really know what is meant by "increased outlet glacier dynamics" because "dynamics" is a general term for changes in glacier speed, thickness and extent. Consider rewording this sentence to clarify your meaning.

Line 20: **utmost**

Line 33/34: I think you can make this point more strongly. It's quite possible a reader could look at Table 1 and this paragraph and think "wow, there are loads of terminus position measurements on the AP", because thousands looks like a lot, then they would be confused when they read this statement on line 33/34. So I think it would help to provide some context along the lines of: "there are approximately 800 tidewater glaciers on the AP [you could count them?], so we are currently missing 800 glaciers x 8 illuminated months x 10 years = 64,000 terminus delineations since 2013 (minus five thousand or so from existing studies), even if we only mapped them once per month, but weekly measurements are

now possible with the abundance of satellite imagery. Plus many glaciers have only ever been measured a handful of times since 1940 (cook et al)", or something to that effect.

Line 34: "we need to use automatic annotation methods". We don't really need to, as demonstrated by the numerous manual delineations on Greenland, but it is much much faster to do it automatically. So consider rephrasing and combining with the following paragraph to emphasise that we now have the tools available to map them automatically.

Line 44: "new reference data": later this is called "training data" are those different or have you switched terminology?

Line 45: This glacier justification is quite weak, but see my major comment above.

Line 49: Sjogren and Boydell were tributaries of Prince Gustav Ice Shelf, not Larsen-A, weren't they?

Line 63: need a comma after "pre-processing"

Line 82: need a comma after "receptive field"

Line 83/84: please specific the threshold for vectorization here and include justification for the choice, and if you add a new figure/section quantifying that impact, it would be good to signpost it here too.

Line 85: I don't quite follow this step because the mask hasn't been described. What is the static mask and how was it derived? What do you do if the glacier retreats or advances beyond the extent of the mask?

Line 89: "separated entries are checked manually" and then put back in if you disagree with the algorithm? Or something else?

Line 105: "more accurate predictions" than what?

Line 123: for glacier modelling, I think the preference would normally be for a raster mask rather than a vector. Consider including masks in addition to the vector dataset, to facilitate use by the modelling community.

Line 130: Without expanding this study to other glaciers, I think a combined analysis of circum-Antarctic calving front change would not be possible, so I'm not sure that this statement is warranted with the current dataset.

Line 134: "such high temporal resolution" is carrying a lot of weight here. Given that the terminus of those glaciers have already been delineated regularly in recent studies (Ochwat et al., 2022; Surawy-Stepney et al., 2023), I don't think this statement is justified.

Line 141: I'm not sure what the purpose of this statement is given that this doesn't appear to be an operational product. Consider removing or rephrasing.

Figure 1: "Larsen Ice Shelf" should be "Larsen-C Ice Shelf", if it even needs to be labelled at all.

Table 2: I'm not a machine learning user, so this may be a stupid question. Should any of the binary classification metrics have units?

Figure 3: I think this would be clearer without the manually digitised terminus on. Or at least it would be nice to see a version like that in the supplementary information.

Benjamin Davison

---

## Author Comment (AC1)

Dear Benjamin Davison,

We thank you for your constructive comments and the careful assessment on our manuscript. All comments have been taken into account and a list of responses and actions is given below. Again, many thanks for helping to improve our manuscript!

Best wishes,

Erik Loebel and all co-authors

**General comments:**

Firstly, I am not an expert in machine or deep learning techniques, and I can see that the underlying method has already been described in detail in Loebel et al. (2022) and applied to 23 Greenland glaciers in Loebel et al. (2023, in review). I will therefore focus my review on the (1) the dataset itself and the aspects of the methodology relevant to producing a time-series of calving front positions for the community and, (2) the accuracy assessment of the method.

The paper presents an exciting application of an existing deep learning method for delineating glacier calving fronts to 19 glaciers on the Antarctic Peninsula. Compared to Greenland, relatively few terminus position datasets are available for the Antarctic Peninsula and the generation of new terminus positions has not kept pace with the generation of new velocity measurements. As a potential future user of this dataset, it is great to see this application and I am confident that new terminus position delineations on the Peninsula will benefit the community. As such, I am wholeheartedly in support of the generation and publication of these datasets. However, I do not yet think that the presented dataset or manuscript meets the quality and scope required for publication in ESSD, but I am hopeful that the authors will take on board my criticisms and suggestions so that this manuscript and dataset can meet the needs of the community and make best use of the deep learning tool that the author has developed.

**Specific comments:**

1a) The scope of the dataset

The authors present a total 2064 calving front delineations across 19 outlet glaciers from 2013 to 2023. One the big questions I had after reading the manuscript was "why not more?". Just to be clear, I don't wish to belittle the efforts of the authors – I am sure it is a lot of work to do this and I know it is a lot of work to generate new datasets. However, there are 1,728 basins in the Cook et al. (2014) basin dataset, roughly half of which terminate in an ice shelf, so there are perhaps 800-odd glaciers on the Peninsula that could be targeted by this method. Since the deep learning method was already developed and the majority of the training dataset already existed, and because comparisons to regions outside of Greenland have already been presented in Loebel et al. (2023), it seems like a relatively small additional contribution to run the processing system for just 19 glaciers, especially given that ESSD does not demand any analysis seeking to develop new understanding from the presented dataset, which is typically the bulk of the work in other journals. Again, I am sure it was a lot of work to do this, which I don't want to detract from, but one of the key benefits of the method used in the

manuscript is that it is automatic and much faster than manual approaches, so it should be able to provide "additional and more comprehensive data products". Therefore, I don't think it is sufficient to present a terminus position dataset that is (for example) ~25% smaller than that in Wallis et al. (2023), given that the dataset in Wallis et al. (2023) was a relatively small component of their publication. It would be great to see a definitive dataset of terminus positions for the Antarctic Peninsula over the last decade – this and the lead author's earlier papers demonstrate that we now have the tools and imagery available to achieve this, so I think that is something we should strive for. In order for this dataset to be suitable for publication in ESSD, and to really demonstrate the utility of the underlying deep learning method, I strongly suggest that it should be applied to many more glaciers on the Antarctic Peninsula.

If there is a good scientific or resource reason for limiting the analysis to a small subset of glaciers, then I would still argue for a larger subset including other major glaciers (e.g. Cadman Glacier, which seems like a major omission here), and I think that more justification for the choice of glaciers should be given. At present, the choice is justified twice in the paper, but only briefly and different reasons are given each time.

Thank you for this assessment. Our data product we created was developed with a limited scope in mind. Related to this, the relatively short manuscript which was originally written as "Brief communication" and was changed after submission on the recommendation of the editor. Nevertheless, we completely agree with your statements. We have this automated processing system, the glaciology community needs more data, so it is reasonable to expect us to provide more data.

We are thankful for your suggestions and we will take this opportunity to significantly increase the scope of our work.

For this we define a solid criteria for glacier selection:
1. Area of interest is limited to the AP ice sheet (*Zwally et al., 2002;* Basins 24, 25, 26, and 27).
2. We process marine-terminating glaciers only. A very large number of AP outlet glaciers drain into ice shelves. Especially the glaciers in basins 24 and 27. Calving fronts of the ice shelves are already covered by IceLines data set (*Baumhoer et al., 2023*).
3. We process all glaciers which are listed in the SCAR Composite Gazetteer of Antarctica (CGA) and have a minimum calving front length of 5 km. This criterion is related to the fact that our processing is not optimized for glacier fronts that are significantly smaller than the 15 km by 15 km input tile size. Processing smaller glacier fronts would require significant processing changes, such as reducing the input tile size and increasing the spatial resolution (probably together with integration of the higher resolution panchromatic band).

Based on these criteria, our data product will include 41 glaciers, more than double the number in the first version. Figure R1 shows the updated overview. The revised version will include all this information on glacier selection. The download of additional satellite data has been completed. We are currently processing the data.

[Figure]

**Figure R1.** *Updated overview map of the northern Antarctic Peninsula and the 41 glaciers included in the data product.*

1b) Filtering of 'raw' terminus positions

One of the main focuses of this paper is that it generates new time-series of terminus positions from an existing method. I was surprised therefore that the manuscript didn't describe much post-processing of the terminus positions in order to make an analysis-ready time-series. The only filtering step I could see is that the authors "separate all entries that have an area difference of more than 1 km2 from the previous and following entries". I don't think that is a sufficiently robust outlier removal technique, especially if you choose to apply this to more glaciers. I suggest that the outlier removal routine should (1) account for the speed of the glacier and time separation between measurements; (2) account for the width of the glacier, because 1 km2 changes might be realistic or not depending on their width, and; (3) account for changes along flowlines or similar, not just width-averaged metrics. The time-series presented in the manuscript look reasonably clean, but I had a quick look at the dataset which showed up some places where I think the outlier identification may not be working, for example Birley glacier on 2022/10/03 has what looks like an unrealistic advance along it's southern branch and Sjogren-Boydell on 2022/03/04 has a ~4 km retreat across what I think is a large section of land. Whichever outlier removal approach is used, the authors should provide details of how many delineations are removed through using it and how that affects the results.

We understand this concern. Filtering the predicted calving fronts is a very important part of generating ANN-based data products. Existing studies either use no filtering or very different solutions with varying effects.

The semi-automated approach we have implemented relies on a time series and manual work, as all problematic delineations are checked manually. The manual checks are fast and more reliable than existing automated filtering solutions. This is particularly important for this contribution as the final data product is the main focus.

That said, we agree with your concerns about our filtering and the fixed 1 km² threshold and we are thankful for your suggestions and recommendations to improve the first filtering step. We decided to integrate the second of your suggested outlier removal techniques. This will be particularly beneficial as we now (with the new glaciers) have a more varied distribution of calving front widths.

In addition to applying this new filtering technique to the 22 new glaciers, we will also apply it to the existing ones. Furthermore, Section 2.1. will be expanded to include more information about this filtering process. This is a great suggestion, thank you very much.

On a related note, I couldn't see any description in the manuscript of partial terminus positions. Does the deep learning approach always provide a full terminus trace? What if the glacier terminus is partially obscured by clouds? Please add detail to the manuscript as necessary.

All the calving fronts traces we provide cover the full calving front. Calving front extractions that do not split the glacier box (and therefore do not result in a time series entry) are automatically discarded. This information will be added to the revised manuscript.

1c) Vectorization of the land/ice probability masks

I couldn't see much justification for the choice of a 0.5 threshold or the impact of that choice of the resulting terminus location. This might be described in the author's earlier papers, but I think ESSD is a suitable place to provide more detail and I think it is relevant to the terminus dataset. Firstly, I think the chosen threshold should be clearly stated in the text, rather than only in the figure (apologies if I just missed it). Secondly, I think there should be a clear quantification, and ideally visualisation, of the impact of that threshold on individual terminus locations and the resulting time-series of area change.

Thank you for bringing this up. The ANN used in our processing performs a land cover classification where each image pixel is classified into either a land/glacier or ocean class. The output of the ANN is a floating point number between zero and one. Zero means that the ANN is confident that the corresponding pixel is ocean and one means that the pixel is glacier/land. In other words, pixels with a predicted value greater than 0.5 are classified by the ANN as glacier/land and pixels with a predicted value less than 0.5 are classified as ocean. The boundary between the two classes, at our threshold of 0.5, is the predicted calving front.

Although it is not really an adjustable parameter in our processing, it is possible to change the threshold. This would shift the final calving front away from the calving front predicted by the ANN. It would also contradict the ground truth data we used to train the ANN.

Figure R2 visualizes the effect of the vectorization thresholds 0.25, 0.5 and 0.75 for two example scenes. The first satellite scene (a-c) is representative of the majority of predictions in which the ANN is very certain where the glacier front is located. Consequently, the threshold value has little influence on the final calving front. When the ANN prediction is less confident, for example for a more fragmented calving front (d-f), there are more pixels where the prediction values are around 0.5. Here, the choice of threshold value can have an impact.

We apologize for not making this clear in the manuscript. Section 2.1. will be expanded to include more information on the prediction mask and its post processing. We think that Figure R2 helps to understand the vectorisation process and therefore plan to include it in the supplement.

[Figure]

*Figure R2. The effect of different vectorisation thresholds within our processing for two example images. Shown are the satellite image (a,d), the corresponding floating-point number prediction mask (b,c) as well as an enlarged spot at the front (c,f).*

1d) Error metric for predicted delineations

Could you provide an error metric for each individual delineation, perhaps by using the spread amongst the 5 models and/or the spread amongst different thresholds? I'm aware that such errors are not provided for manual delineations because it is impractical, but it seems achievable and useful for this method, especially given the apparent differences between each of the 5 models on challenging images shown in Figure 3.

This is a very nice idea and we are happy to implement it. We will also add a short description of this new metric to the revised version of our manuscript.

1e) Dataset format

I think that dataset would be much easier and quicker to use if there was just one shapefile for each glacier plus one shapefile containing all delineations for all glaciers. Some users are now also using the geopackage format, so the authors might want to consider providing the output in that format also.

We will also be making our data available in Geopackage format.

1f) Dataset contents

It would be great if the training and test data were also released, along with the automatic delineations. Some glaciers and times seem to be missing a 'coastline' shapefile. Is that expected? Also, the justification in the manuscript for providing two outputs is not clear. Why is the coastline file better than the glacier file for merging with an ice mask? It's implied that the terminus file contains only the glacier edge, whereas the coastline file contains the glacier edge and the edge of the surrounding fjord walls, but this isn't demonstrated clearly nor how that distinction is made if part of the terminus is obscured by clouds, for example. Please add some clarification and justification in that respect to the manuscript.

We fully agree with the recommendation to publish our reference data. The data will be submitted to the TU Dresden *Open Access Repository and Archive* (OpARA). Reference (with doi) will be included in the revised version.

Our ANN predicts the boundary between an ocean and a glacier/land class. For many glaciers, this includes not only the calving front, but also the non-glaciated coastline. For these glaciers, we mask out the calving front from the whole coastline prediction.

For glaciers where this is the case, we have decided to publish the coastline along the calving front. Previous collaboration with ice sheet modelers has shown that these coastline files are useful when integrating calving fronts into an existing ice mask, as the overlap reduces (usually eliminates) the need for interpolation. In any case, we think these coastline files may be useful to some users, and we don't see any harm in publishing them along with the main calving front product.

The current manuscript misses this information. Additional information and clarification on the coastline files will be added to the revised version.

2a) Accuracy assessment

If I have understood it corrected, the accuracy assessment consists of a comparison between the 5-model mean delineation (with a single threshold) and three manual delineations per glacier outside of the training window. Only one of those images for 10 of the 19 glaciers is shown and otherwise we are provided with some simple metrics summarising the results of 19x3 comparisons. In my view, that is not sufficient to characterise the accuracy of the model in this region. Figure 3 is a useful illustration (though I have some suggestions below), but there is no evidence given that demonstrates that the examples given in Figure 3 are representative of the typical accuracy of the method under those conditions, or what the spread in performance is like in each of those conditions. I suggest that the accuracy assessment should include enough images to provide statistically significant accuracy measures of accuracy for glaciers and images with each of the different characteristics shown in

Figure 3. Ideally, it would also show the effect of combinations of those conditions, such as times of low illumination with a scene border, with or without mélange and cloud cover.

Figure 3: this is one of the main pieces of evidence presented in the manuscript to convince the reader that the automatic delineations performs as well as a manual delineation. However, showing ~15x15 km images to illustrate differences in position of less than 100 m is not a very clear way to illustrate those differences – the figures would need to be produced at an impractical resolution and I would need a much better monitor to see anything meaningful, and even then I wouldn't be able to measure the differences. I suggest that an additional, more quantitative figure should be provided, to show differences between the automatic and manual delineations for the full test dataset. Perhaps some simplegraphs with 'distance along terminus' on the x-axis and 'difference from manual delineation' on the y-axis would allow the authors to plot the differences through the full test dataset every 30 m along each delineation? That kind of plot would also clearly show how those differences are affected by your choice of model and threshold for vectorization. You could have one graph per glacier and perhaps show a histogram for each glacier.

Thank you for raising this very important point. Figure 3 was created to demonstrate that the ANN is able to find the calving front in different image conditions and not necessarily to highlight and understand the manual delineation differences of less than 100 meters. Main argument for the performance of the model is given in Table 2 (and Table R1).

However, we are thankful for the suggestion and we are happy to extend Figure 3 to give more evidence for the performance of our method. As you mentioned, the difference to manual delineation is not uniform along the calving fronts and changes from test image to test image. This is currently not shown in the manuscript.

For the revised version, Figure 3 will include the proposed graphs and histograms showing the differences to manual delineation along the predicted front (for each model). These plots will also link well to our new Hausdorff distance error metric introduced in Table R1.

In addition to the updated Figure 3, we will create another figure showing a histogram of the differences from manual delineation for the entire test dataset. This figure emphasizes the non-normal distribution of the test results.

I am unsure that the accuracy metrics of mean and median difference from manual delineations is representative of the differences between the automatic and manual delineations with regard to evaluating the use of the automatic delineations for scientific purposes. As far as I can tell, both of those metrics would be insensitive to large differences between automatic and manual delineations if those differences occur over a short section of the terminus. For example, Figure 3 shows automatic delineations on Prospect are in several places over 1 km from the manual delineation, but the mean difference is small because there are comparatively long sections where the two sets of delineations are in close agreement. This shows up a bit in the Prospect timeseries in Figure 4g, where there is a ~10 km2 difference between the automatic and manual delineation in late-2022. For glaciological and modelling applications, it might be that those areas of large difference are the bits that matter, so the mean error across the terminus wouldn't be a useful error metric. The other problem with this is that the mean or median difference between the delineations will be highly dependent the length of glacier terminus compared to the length of non-glacier digitised coastline.

*The differences between automated and manual delineation are not evenly distributed along the glacier front. Although we are of the opinion that the mean difference we report is somewhat sensitive to regions with larger differences (the aforementioned test image of Prospect Glacier has a mean difference of 123 meters due to the difficult section, whereas the other test images often have a difference of less than 30 meters), we understand this concern.*

*Therefore, in addition to the difference estimate (which we will now call average minimal distance), we would like to introduce a further error metric, the Hausdorff distance (Huttenlocher et al., 1993). The Hausdorff distance considers only the largest of all minimum distances along the two trajectories and is therefore highly sensitive towards misclassified parts along the calving front. Taken together, these two very different distance estimates will allow a better categorisation of our results. Table R1 shows the accuracy assessment results for our test set. Descriptions will be added to the revised manuscript.*

**Table R1.** *Extended table with the accuracy assessment result. In addition to the average minimal distance and the binary classification metrics (see Table 2, Preprint), the Hausdorff distance is also included.*

| Average minimal distance | | Hausdorff distance | | Binary classification metrics | | | |
|---|---|---|---|---|---|---|---|
| Mean (m) | Median (m) | Mean (m) | Median (m) | Accuracy | Precision | Recall | F1-Score |
| $59.3 \pm 5.9$ | $33.9 \pm 1.5$ | $405.1 \pm 20.7$ | $257.0 \pm 14.7$ | $0.984 \pm 0.001$ | $0.978 \pm 0.002$ | $0.995 \pm 0.001$ | $0.986 \pm 0.001$ |

As presented, there isn't a compelling demonstration that the dataset is as applicable to science cases than manually-derived datasets, which I think is important given the proposed justification for making the dataset. Another more holistic approach the authors should take to demonstrate the quality of their time-series product, which would go a long way to addressing that concern, would be to compare time series of area change from these new delineations to area change time-series derived from other terminus position datasets, where both/multiple datasets have sampled the same glacier during overlapping time periods.

*We also appreciate this suggestion and are happy to implement it. Our updated time series will be compared to other manually delineated data products (see Table 1) for representative examples. This will further validate our results but also show differences in sampling rate.*

3) Introduction

I do not think the introduction adequately justifies the need for improved monitoring of outlet glacier terminus position change. As written, it states that (1) ice shelves have reduced in thickness and extent, which has led to glacier speed-up, (2) calving fronts can be used to study ice-ocean interaction and that (3) they can be used to improve model simulations. Those points are all true, but I think they need more detail and specifics in order to make a convincing argument for this new dataset. Consider including more detail of ice shelf and glacier area changes (citing the various papers by Cook et al on the subject) and how much the Peninsula has contributed to Antarctica's total sea level contribution. Are there specific examples of where model performance has been limited or improved by the availability of calving front positions, or has it been quantified in a more general sense? For such

models, I think they would they need a continual coastline across the whole domain, not small subsets as provided here. In addition, consider drawing on the literature from Greenland, where measurements of terminus position change have led improvements in our understanding of glacier response to environmental conditions over a range of spatial and temporal scales (e.g. Cowton et al., 2018) or have at least aided the interpretation of changes in ice speed, and how they have been used in combination with estimates of submarine melt rates to develop new parameterisations for the impact of submarine melting on calving and terminus position (Slater et al., 2019). Terminus positions are really useful, but I don't think that comes across in the introduction as currently written.

Thank you for your comments. Now that we are not limited by space, we are happy to extend the introduction and include your suggestions. In particular we will go into more detail describing glacier and ice shelf area changes and why we need them, we will include numbers of AP ice mass loss and sea level contribution and we will document that calving fronts are an important constraint for ice dynamic modeling and thus improve simulations of future mass loss and sea level contribution. For this we will also include references from Greenland (like *Vieli and Nick, 2011* or *Bondizo et al., 2017*). Thank you for these suggestions.

Cowton, T.R., Sole, A.J., Nienow, P.W., Slater, D.A. and Christoffersen, P., 2018. Linear response of east Greenland's tidewater glaciers to ocean/atmosphere warming. Proceedings of the National Academy of Sciences, 115(31), pp.7907-7912.

Slater, D. A., Straneo, F., Felikson, D., Little, C. M., Goelzer, H., Fettweis, X., and Holte, J.: Estimating
Greenland tidewater glacier retreat driven by submarine melting, The Cryosphere, 13, 2489–2509, https://doi.org/10.5194/tc-13-2489-2019, 2019.

Huttenlocher, D., Klanderman, G., and Rucklidge, W.: Comparing images using the Hausdorff distance, IEEE Transactions on Pattern Analysis and Machine Intelligence, 15, 850–863, https://doi.org/10.1109/34.232073, 1993.

Bondizo, J. H., Morlighem, M., Seroussi, H., Kleiner, T., Rückamp, M., Mouginot, J., Moon, T., Larour, E. Y., and Humbert, A.: The mechanisms behind Jakobshavn Isbræ's acceleration and mass loss: A 3-D thermomechanical model study, Geophysical Research Letters, 44, 6252–6260, https://doi.org/10.1002/2017GL073309, 2017.

Vieli, A. and Nick, F. M.: Understanding and Modelling Rapid Dynamic Changes of Tidewater Outlet Glaciers: Issues and Implications, Surveys in Geophysics volume, 32, 437 – 458, https://doi.org/10.1007/s10712-011-9132-4, 2011.

**Minor comments**

Line 4: suggest "rely on manual delineation, **which is** time-consuming"

Will be fixed, thank you.

Line 8: suggest "The data product presented here"

We will follow this suggestion.

Line 16: ice **shelf**

Will be fixed.

Line 17: "forcing from ocean and atmosphere has led to reduced ice shelf thickness and extent. And this, in turn, has reduced buttressing strength and thereby increased outlet glacier dynamics". I don't think this is a fair summary of our current understanding of ice shelf and glacier changes on the Peninsula. Can you add more detail on what is meant by "forcing". I don't really know what is meant by "increased outlet glacier dynamics" because "dynamics" is a general term for changes in glacier speed, thickness and extent. Consider rewording this sentence to clarify your meaning.

More detail will be added to the introduction. This statement will be clarified.

Line 20: **utmost**

Will be fixed.

Line 33/34: I think you can make this point more strongly. It's quite possible a reader could look at Table 1 and this paragraph and think "wow, there are loads of terminus position measurements on the AP", because thousands looks like a lot, then they would be confused when they read this statement on line 33/34. So I think it would help to provide some context along the lines of: "there are approximately 800 tidewater glaciers on the AP [you could count them?], so we are currently missing 800 glaciers x 8 illuminated months x 10 years = 64,000 terminus delineations since 2013 (minus five thousand or so from existing studies), even if we only mapped them once per month, but weekly measurements are now possible with the abundance of satellite imagery. Plus many glaciers have only ever been measured a handful of times since 1940 (cook et al)", or something to that effect.

Thank you for your thoughts and suggestions. We will take them into account when we revise the introduction.

Line 34: "we need to use automatic annotation methods". We don't really need to, as demonstrated by the numerous manual delineations on Greenland, but it is much much faster to do it automatically. So consider rephrasing and combining with the following paragraph to emphasise that we now have the tools available to map them automatically.

This will be rephrased, thanks.

Line 44: "new reference data": later this is called "training data" are those different or have you switched terminology?

In our manuscript, we use the term *reference data* to describe all (manually) labeled data. The reference data is divided into *training data* (images from 2013 to 2021) and *test data* (images from 2022 and 2023). The description of the reference data set in Section 2 will be expanded to clarify this.

Line 45: This glacier justification is quite weak, but see my major comment above.

We now have a solid set of criteria for selecting glaciers. See our answer above (page 2).

Line 49: Sjogren and Boydell were tributaries of Prince Gustav Ice Shelf, not Larsen-A, weren't they?

That is correct, we will change that. Thank you very much.

Line 63: need a comma after "pre-processing"

Will be fixed.

Line 82: need a comma after "receptive field"

Will be fixed.

Line 83/84: please specific the threshold for vectorization here and include justification for the choice, and if you add a new figure/section quantifying that impact, it would be good to signpost it here too.

Please see our answer above (page 4).

Line 85: I don't quite follow this step because the mask hasn't been described. What is the static mask and how was it derived? What do you do if the glacier retreats or advances beyond the extent of the mask?

These static masks are created for each glacier and cover the entire input image. Section 2.1. will be expanded to include more information on the post processing of our method.

Line 89: "separated entries are checked manually" and then put back in if you disagree with the algorithm? Or something else?

Yes, entries which have been separated due to true area change (e.g. due to calving of an iceberg) are reinserted into the data set. Entries which have been separated due to a misclassification by the ANN are discarded.

Together with the description of the new filtering approach (see page 4), we will also expand the description of this procedure. Thank you for your question.

Line 105: "more accurate predictions" than what?

More accurate than single band inputs. This will be clarified.

Line 123: for glacier modelling, I think the preference would normally be for a raster mask rather than a vector. Consider including masks in addition to the vector dataset, to facilitate use by the modelling community.

This was also raised by referee 2. We will include polygonal masks in the revised version.

Line 130: Without expanding this study to other glaciers, I think a combined analysis of circum-Antarctic calving front change would not be possible, so I'm not sure that this statement is warranted with the current dataset.

More glaciers will be included.

Line 134: "such high temporal resolution" is carrying a lot of weight here. Given that the terminus of

those glaciers have already been delineated regularly in recent studies (Ochwat et al., 2022; Surawy-Stepney et al., 2023), I don't think this statement is justified.

Thank you for this comment and the references. We will review this statement and change it accordingly.

Line 141: I'm not sure what the purpose of this statement is given that this doesn't appear to be an operational product. Consider removing or rephrasing.

You are right, this will be removed.

Figure 1: "Larsen Ice Shelf" should be "Larsen-C Ice Shelf", if it even needs to be labelled at all.

The label will be removed in the updated figure.

Table 2: I'm not a machine learning user, so this may be a stupid question. Should any of the binary classification metrics have units?

The four binary classification metrics we used have no unit. However, we will add to the description that they can be interpreted as percentages.

Figure 3: I think this would be clearer without the manually digitised terminus on. Or at least it would be nice to see a version like that in the supplementary information.

We will try to make this figure clearer (using transparency and different marker sizes). If this does not work out we will include a version without the manually digitized terminus in the supplement.

Benjamin Davis

---

## Author Comment (AC2)

Dear Referee #2,

Thank you for your review and for the constructive feedback on our manuscript. All comments have been taken into account and a list of responses and actions is given below.

Best wishes,

Erik Loebel and all co-authors

**General Comments:**

This work consists of a automatically generated glacial termini data product for 19 key outlet glaciers along the Antarctica Peninsula, and includes 2064 calving front locations from 2013 to 2023 at sub-seasonal temporal resolutions.

The manuscript covers the current state of the art in the field of machine learning/deep learning based cryosphere data extraction methods, as well as the need for such methods to be applied towards glacial data extraction. It describes the importance of calving front data for understanding dynamic glacier changes of marine terminating glaciers, and improving ice modeling. To address the labor-intensive obstacles required by manual delineation, an automatic deep learning-based processing system is developed to extract glacier fronts from satellite imagery. By leveraging the generalization capability of machine learning techniques to provide new observational constraints, this study contributes to groundwork that will enhance the cryosphere community's understanding of glacial dynamics and ice-ocean interactions.

The method uses the deep neural network trained on existing datasets to process Landsat 8 & 9 multiband imagery at spatial resolutions of 30m, and output Shapefile polylines at a spatial accuracy of 59.3 ± 5.9 m (average distance between the measured and predicted fronts). This which falls within human levels of accuracy (<107 m, Goliber et al. 2022).

The dataset itself is composed of zip files of the 19 basins, which are further organized into folders for each observed date, which then contain 2 set of Shapefiles (1 for the coastline, and 1 for the extracted glacial front). Metadata provides the name and date of the processed front. While the scope is small, the dataset still provides valuable new observational constraints.

We are thankful for your comments and remarks. In terms of the scope of our work, we'd like to inform you of planned changes to this data product (see RC1, Pages 1-3), in particular the increase in the number of glaciers covered from 19 to 41.

The publication is well done, and is largely free of grammatical errors and typographical issues. There are minor remarks to be addressed by the authors, after which I can recommend acceptance at the editor's discretion.

**Specific Comments:**

Dataset Coastline Quality

While the majority of the dataset is well curated, there are some coastlines in the dataset (i.e., drygalski_20210301_coastline, murphy_wilkinson_20191114_coastline, cayley_20141015_coastline,

cayley_20200227_coastline) that seem to have erroneous delineations, particularly along the domain boundaries. More validation or pruning of these data is needed, i.e. by manual pruning through visual GIS software, or some automated pruning by checking inter-annual differences between fronts to detect outliers.

The entire dataset will be reprocessed (see RC1, Pages 3-4) using the new width-dependent filtering. We will also manually check the data, especially the example you mentioned.

Dataset Coastlines Polygons

In conjunction with the above comment, it would be useful to have the glacial termini data in the form of land/ocean polygonal masks in addition to just a polyline, though this may be outside the scope of this work. This would also resolve the errors along the domain boundaries. Alternatively, provision of the domain boundaries would be helpful, as this would make it easier for modelers/community members to judge where errors are, and/or where the coastlines can be stitched to existing land/ocean masks.

Thank you for this suggestion. This was also raised by Referee #1. We are happy to include polygonal masks.

Dataset format

The organization of the dataset could be streamlined, such that the user can load an entire time series of a single domain without having to enter/navigate individual folders for each date, and/or make it more manageable for GIS software on less capable machines to load in all at once. Alternatively, such shapefiles could be consolidated, and the ability of Shapefiles to hold multiple features/delineations within a single file would be of use. Provision of monthly, quarterly, annual, or full time series files (similar to IceLines, Baumhoer et al., 2023) should be within scope.

The revised version will include a consolidated file for each glacier with all calving front traces.We will also include monthly and annual files for each glacier. Thanks for these suggestions.

Full time series results (area changes) are already available (http://dx.doi.org/10.25532/OPARA-277). These will be updated to include the new glaciers.

Accuracy Comparison w.r.t. Other Datasets

The mean/median distance and binary classification metrics are established accuracy measures in the calving front delineation field, and this study performs well on the evaluated test set. Considering L113P6: ("Although completely different test data sets are involved…Loebel et al. (2023c)."), it may be within scope to see a comparison with existing test sets/studies, to ensure the chosen test set is not biased, and the accuracy metrics are comparable. That being said, the generalization of the network is recognizable, so this can be done at the author's discretion.

As our model uses multi-spectral inputs, there are currently no test or benchmark datasets other than the one we have processed. We can not test the model applied for this study with our TUD test set of *Loebel et al. 2023* as these test images are now included in the training data. However we can apply our model to the CALFIN and ESA-CCI test sets used in *Loebel et al. 2023*. The two test datasets

contain an additional 100 and 110 test images of Greenland outlet glaciers. Results will be reported in the revised version together with an extended accuracy assessment table in the supplement.

In conjunction with these additions, we'd like to let you know about planned changes to our accuracy assessment (see RC1, Pages 6-8).

**Minor comments:**

- It would be helpful to provide the spatial accuracy of the data to readers in the abstract.

Spatial accuracy will be included in the abstract.

---

## Author Response (AR1)

Dear Baptiste Vandecrux,

The additions to our data product and the revision of the manuscript took longer than expected. We would like to thank you for extending the deadline.
We have uploaded this revised manuscript together with a track change version. We have carefully addressed the points raised by the referees and made changes in line with our responses. Below, we also provide definitive answers to how and where the referee's comments were addressed. For comprehensive answers to the questions, please see our responses (Reply on RC1/2).

In particular, the following four major changes were made to the dataset and manuscript.
- We have significantly increased the coverage of our data product from 19 to 42 glaciers. The updated data product is now available on PANGAEA (same doi).
- The introduction has been significantly revised. This is in line with Benjamin Davison's (RC1) comments and our responses.
- The accuracy assessment of our data has been extended. The biggest changes are the introduction of the Hausdorff distance estimate and the inter-model distance. The inter-model distance is now also included in the attribute table for each extracted calving front.
- Section 3 now includes a comparison with existing data products for three examples. This helps to contextualize our work.

Thank you for the work on our Manuscript. Best wishes,

Erik and all co-authors

**Benjamin Davison, 28 Feb 2024**

**General comments:**

Firstly, I am not an expert in machine or deep learning techniques, and I can see that the underlying method has already been described in detail in Loebel et al. (2022) and applied to 23 Greenland glaciers in Loebel et al. (2023, in review). I will therefore focus my review on the (1) the dataset itself and the aspects of the methodology relevant to producing a time-series of calving front positions for the community and, (2) the accuracy assessment of the method.

The paper presents an exciting application of an existing deep learning method for delineating glacier calving fronts to 19 glaciers on the Antarctic Peninsula. Compared to Greenland, relatively few terminus position datasets are available for the Antarctic Peninsula and the generation of new terminus positions has not kept pace with the generation of new velocity measurements. As a potential future user of this dataset, it is great to see this application and I am confident that new terminus position delineations on the Peninsula will benefit the community. As such, I am wholeheartedly in support of the generation and publication of these datasets. However, I do not yet think that the presented dataset or manuscript meets the quality and scope required for publication in ESSD, but I am hopeful that the authors will take on board my criticisms and suggestions so that this manuscript and dataset can meet the needs of the community and make best use of the deep learning tool that the author has developed.

**Specific comments:**

1a) The scope of the dataset

The authors present a total 2064 calving front delineations across 19 outlet glaciers from 2013 to 2023. One the big questions I had after reading the manuscript was "why not more?". Just to be clear, I don't wish to belittle the efforts of the authors – I am sure it is a lot of work to do this and I know it is a lot of work to generate new datasets. However, there are 1,728 basins in the Cook et al. (2014) basin dataset, roughly half of which terminate in an ice shelf, so there are perhaps 800-odd glaciers on the Peninsula that could be targeted by this method. Since the deep learning method was already developed and the majority of the training dataset already existed, and because comparisons to regions outside of Greenland have already been presented in Loebel et al. (2023), it seems like a relatively small additional contribution to run the processing system for just 19 glaciers, especially given that ESSD does not demand any analysis seeking to develop new understanding from the presented dataset, which is typically the bulk of the work in other journals. Again, I am sure it was a lot of work to do this, which I don't want to detract from, but one of the key benefits of the method used in the manuscript is that it is automatic and much faster than manual approaches, so it should be able to provide "additional and more comprehensive data products". Therefore, I don't think it is sufficient to present a terminus position dataset that is (for example) ~25% smaller than that in Wallis et al. (2023), given that the dataset in Wallis et al. (2023) was a relatively small component of their publication. It would be great to see a definitive dataset of terminus positions for the Antarctic Peninsula over the last decade – this and the lead author's earlier papers demonstrate that we now have the tools and imagery available to achieve this, so I think that is something we should strive for. In order for this dataset to be suitable for publication in ESSD, and to really demonstrate the utility of the underlying deep learning method, I strongly suggest that it should be applied to many more glaciers on the Antarctic Peninsula.

If there is a good scientific or resource reason for limiting the analysis to a small subset of glaciers, then I would still argue for a larger subset including other major glaciers (e.g. Cadman Glacier, which seems like a major omission here), and I think that more justification for the choice of glaciers should be given. At present, the choice is justified twice in the paper, but only briefly and different reasons are given each time.

We have significantly increased the scope of our data product. It now includes 42 glaciers (4817 calving front traces) instead of the original 19 glaciers (2064 traces). Much of the text and figures have been modified or expanded.

Please find the updated data product here (same doi als the first submission): https://doi.pangaea.de/10.1594/PANGAEA.963725

For the selection of these 42 glaciers we defined solid criteria (see P3 L67).

1b) Filtering of 'raw' terminus positions
One of the main focuses of this paper is that it generates new time-series of terminus positions from an existing method. I was surprised therefore that the manuscript didn't describe much post-processing of the terminus positions in order to make an analysis-ready time-series. The only filtering step I could see is that the authors "separate all entries that have an area difference of more than 1 km2 from the previous and following entries". I don't think that is a sufficiently robust outlier

removal technique, especially if you choose to apply this to more glaciers. I suggest that the outlier removal routine should (1) account for the speed of the glacier and time separation between measurements; (2) account for the width of the glacier, because 1 km2 changes might be realistic or not depending on their width, and; (3) account for changes along flowlines or similar, not just width-averaged metrics. The time-series presented in the manuscript look reasonably clean, but I had a quick look at the dataset which showed up some places where I think the outlier identification may not be working, for example Birley glacier on 2022/10/03 has what looks like an unrealistic advance along it's southern branch and Sjogren-Boydell on 2022/03/04 has a ~4 km retreat across what I think is a large section of land. Whichever outlier removal approach is used, the authors should provide details of how many delineations are removed through using it and how that affects the results.

We have applied the suggested filtering technique (second suggestion) to all 42 glaciers. Section 2.1 has been expanded to include all relevant information (P5 L106). Figure 2 has been expanded to include more details on quality control.

Furthermore we provided more insights into the statistics of the filtering (P6 L113).

On a related note, I couldn't see any description in the manuscript of partial terminus positions. Does the deep learning approach always provide a full terminus trace? What if the glacier terminus is partially obscured by clouds? Please add detail to the manuscript as necessary.

This information has been included in the revised manuscript (P6 L119).

1c) Vectorization of the land/ice probability masks

I couldn't see much justification for the choice of a 0.5 threshold or the impact of that choice of the resulting terminus location. This might be described in the author's earlier papers, but I think ESSD is a suitable place to provide more detail and I think it is relevant to the terminus dataset. Firstly, I think the chosen threshold should be clearly stated in the text, rather than only in the figure (apologies if I just missed it). Secondly, I think there should be a clear quantification, and ideally visualisation, of the impact of that threshold on individual terminus locations and the resulting time-series of area change.

See our response (https://doi.org/10.5194/essd-2023-535-AC1).

We have included additional explanations of the vectorisation process (P5 L96). We have also added a new figure to the Supplement (Fig. S2).

1d) Error metric for predicted delineations

Could you provide an error metric for each individual delineation, perhaps by using the spread amongst the 5 models and/or the spread amongst different thresholds? I'm aware that such errors are not provided for manual delineations because it is impractical, but it seems achievable and useful for this method, especially given the apparent differences between each of the 5 models on challenging images shown in Figure 3.

We have included this new metric under the name "inter-model distance". Each calving front trace in our data product now has a corresponding estimate. The metric and its implementation are explained in the new section 2.2.2.

1e) Dataset format

I think that dataset would be much easier and quicker to use if there was just one shapefile for each glacier plus one shapefile containing all delineations for all glaciers. Some users are now also using the geopackage format, so the authors might want to consider providing the output in that format also.

All data is now available in geopackage format in addition to shapefile format.

1f) Dataset contents

It would be great if the training and test data were also released, along with the automatic delineations. Some glaciers and times seem to be missing a 'coastline' shapefile. Is that expected? Also, the justification in the manuscript for providing two outputs is not clear. Why is the coastline file better than the glacier file for merging with an ice mask? It's implied that the terminus file contains only the glacier edge, whereas the coastline file contains the glacier edge and the edge of the surrounding fjord walls, but this isn't demonstrated clearly nor how that distinction is made if part of the terminus is obscured by clouds, for example. Please add some clarification and justification in that respect to the manuscript.

All reference data used in this study (both Greenland and AP glaciers) are now publicly available (with president doi). See section 4 for details.

We have added additional information on processing and coastline files (P5 L100). Coastlines files are now only available for the new annual product. There are two reasons for this: Firstly, coastlines are almost static and do not benefit from sub-seasonal sampling. Secondly, as our quality control focuses only on the calving front, we cannot guarantee a high quality of all coastline extractions (all annual entries are quality controlled manually, see answers for RC2 below).

2a) Accuracy assessment

If I have understood it corrected, the accuracy assessment consists of a comparison between the 5-model mean delineation (with a single threshold) and three manual delineations per glacier outside of the training window. Only one of those images for 10 of the 19 glaciers is shown and otherwise we are provided with some simple metrics summarising the results of 19x3 comparisons. In my view, that is not sufficient to characterise the accuracy of the model in this region. Figure 3 is a useful illustration (though I have some suggestions below), but there is no evidence given that demonstrates that the examples given in Figure 3 are representative of the typical accuracy of the method under those conditions, or what the spread in performance is like in each of those conditions. I suggest that the accuracy assessment should include enough images to provide statistically significant accuracy measures of accuracy for glaciers and images with each of the different characteristics shown in Figure 3. Ideally, it would also show the effect of combinations of those conditions, such as times of low illumination with a scene border, with or without mélange and cloud cover.

Figure 3: this is one of the main pieces of evidence presented in the manuscript to convince the reader that the automatic delineations performs as well as a manual delineation. However, showing ~15x15 km images to illustrate differences in position of less than 100 m is not a very clear way to illustrate

those differences – the figures would need to be produced at an impractical resolution and I would need a much better monitor to see anything meaningful, and even then I wouldn't be able to measure the differences. I suggest that an additional, more quantitative figure should be provided, to show differences between the automatic and manual delineations for the full test dataset. Perhaps some simplegraphs with 'distance along terminus' on the x-axis and 'difference from manual delineation' on the y-axis would allow the authors to plot the differences through the full test dataset every 30 m along each delineation? That kind of plot would also clearly show how those differences are affected by your choice of model and threshold for vectorization. You could have one graph per glacier and perhaps show a histogram for each glacier.

The accuracy assessment section (section 2.2) has been revised. There is an updated Figure 3 showing the distance error along the calving front trajectories and a new Figure 4 showing the histogram of the average minimal distance over the entire test data. Figure 5 shows the mean inter-model distance histogram over all data product entries (Figs. S3 and S4 separately for each glacier).

I am unsure that the accuracy metrics of mean and median difference from manual delineations is representative of the differences between the automatic and manual delineations with regard to evaluating the use of the automatic delineations for scientific purposes. As far as I can tell, both of those metrics would be insensitive to large differences between automatic and manual delineations if those differences occur over a short section of the terminus. For example, Figure 3 shows automatic delineations on Prospect are in several places over 1 km from the manual delineation, but the mean difference is small because there are comparatively long sections where the two sets of delineations are in close agreement. This shows up a bit in the Prospect timeseries in Figure 4g, where there is a ~10 km2 difference between the automatic and manual delineation in late-2022. For glaciological and modelling applications, it might be that those areas of large difference are the bits that matter, so the mean error across the terminus wouldn't be a useful error metric. The other problem with this is that the mean or median difference between the delineations will be highly dependent the length of glacier terminus compared to the length of non-glacier digitised coastline.

We included the Hausdorff distance estimate in our accuracy assessment in Section 2.2.1.

As presented, there isn't a compelling demonstration that the dataset is as applicable to science cases than manually-derived datasets, which I think is important given the proposed justification for making the dataset. Another more holistic approach the authors should take to demonstrate the quality of their time-series product, which would go a long way to addressing that concern, would be to compare time series of area change from these new delineations to area change time-series derived from other terminus position datasets, where both/multiple datasets have sampled the same glacier during overlapping time periods.

We compared our dataset with existing time series for three examples. See Section 3 (from P11 L222) and Figure 7.

3) Introduction

I do not think the introduction adequately justifies the need for improved monitoring of outlet glacier terminus position change. As written, it states that (1) ice shelves have reduced in thickness and extent, which has led to glacier speed-up, (2) calving fronts can be used to study ice-ocean interaction and that (3) they can be used to improve model simulations. Those points are all true, but I think they

need more detail and specifics in order to make a convincing argument for this new dataset. Consider including more detail of ice shelf and glacier area changes (citing the various papers by Cook et al on the subject) and how much the Peninsula has contributed to Antarctica's total sea level contribution. Are there specific examples of where model performance has been limited or improved by the availability of calving front positions, or has it been quantified in a more general sense? For such models, I think they would they need a continual coastline across the whole domain, not small subsets as provided here. In addition, consider drawing on the literature from Greenland, where measurements of terminus position change have led improvements in our understanding of glacier response to environmental conditions over a range of spatial and temporal scales (e.g. Cowton et al., 2018) or have at least aided the interpretation of changes in ice speed, and how they have been used in combination with estimates of submarine melt rates to develop new parameterisations for the impact of submarine melting on calving and terminus position (Slater et al., 2019). Terminus positions are really useful, but I don't think that comes across in the introduction as currently written.

The introduction has been significantly expanded to include more details on AP ice mass loss, changes in ice shelf area, and the use of calving fronts in ice dynamics modeling.

Cowton, T.R., Sole, A.J., Nienow, P.W., Slater, D.A. and Christoffersen, P., 2018. Linear response of east Greenland's tidewater glaciers to ocean/atmosphere warming. Proceedings of the National Academy of Sciences, 115(31), pp.7907-7912.

Slater, D. A., Straneo, F., Felikson, D., Little, C. M., Goelzer, H., Fettweis, X., and Holte, J.: Estimating
Greenland tidewater glacier retreat driven by submarine melting, The Cryosphere, 13, 2489–2509, https://doi.org/10.5194/tc-13-2489-2019, 2019.

**Minor comments**

Line 4: suggest "rely on manual delineation, **which is** time-consuming"

Fixed

Line 8: suggest "The data product presented here"

Done

Line 16: ice **shelf**

Fixed

Line 17: "forcing from ocean and atmosphere has led to reduced ice shelf thickness and extent. And this, in turn, has reduced buttressing strength and thereby increased outlet glacier dynamics". I don't think this is a fair summary of our current understanding of ice shelf and glacier changes on the Peninsula. Can you add more detail on what is meant by "forcing". I don't really know what is meant by "increased outlet glacier dynamics" because "dynamics" is a general term for changes in glacier speed, thickness and extent. Consider rewording this sentence to clarify your meaning.

Introduction has been overhauled

Line 20: **utmost**

Fixed

Line 33/34: I think you can make this point more strongly. It's quite possible a reader could look at Table 1 and this paragraph and think "wow, there are loads of terminus position measurements on the AP", because thousands looks like a lot, then they would be confused when they read this statement on line 33/34. So I think it would help to provide some context along the lines of: "there are approximately 800 tidewater glaciers on the AP [you could count them?], so we are currently missing 800 glaciers x 8 illuminated months x 10 years = 64,000 terminus delineations since 2013 (minus five thousand or so from existing studies), even if we only mapped them once per month, but weekly measurements are now possible with the abundance of satellite imagery. Plus many glaciers have only ever been measured a handful of times since 1940 (cook et al)", or something to that effect.

Introduction has been overhauled

Line 34: "we need to use automatic annotation methods". We don't really need to, as demonstrated by the numerous manual delineations on Greenland, but it is much much faster to do it automatically. So consider rephrasing and combining with the following paragraph to emphasise that we now have the tools available to map them automatically.

Rephrased

Line 44: "new reference data": later this is called "training data" are those different or have you switched terminology?

Clarified in section 2.1

Line 45: This glacier justification is quite weak, but see my major comment above.

We now have solid criteria (see P3 L67).

Line 49: Sjogren and Boydell were tributaries of Prince Gustav Ice Shelf, not Larsen-A, weren't they?

Fixed

Line 63: need a comma after "pre-processing"

Fixed

Line 82: need a comma after "receptive field"

Fixed

Line 83/84: please specific the threshold for vectorization here and include justification for the choice, and if you add a new figure/section quantifying that impact, it would be good to signpost it here too.

Clarified in Section 2.1 (P5 L96)

Line 85: I don't quite follow this step because the mask hasn't been described. What is the static mask

and how was it derived? What do you do if the glacier retreats or advances beyond the extent of the mask?

Clarified in Section 2.1 (P5 L101)

Line 89: "separated entries are checked manually" and then put back in if you disagree with the algorithm? Or something else?

Clarified in Section 2.1 (P6 L110)

Line 105: "more accurate predictions" than what?

Clarified (P7 L147)

Line 123: for glacier modelling, I think the preference would normally be for a raster mask rather than a vector. Consider including masks in addition to the vector dataset, to facilitate use by the modelling community.

Polygonal masks are included for all coastline predictions.

Line 130: Without expanding this study to other glaciers, I think a combined analysis of circum-Antarctic calving front change would not be possible, so I'm not sure that this statement is warranted with the current dataset.

More glaciers are now included.

Line 134: "such high temporal resolution" is carrying a lot of weight here. Given that the terminus of those glaciers have already been delineated regularly in recent studies (Ochwat et al., 2022; Surawy-Stepney et al., 2023), I don't think this statement is justified.

Removed

Line 141: I'm not sure what the purpose of this statement is given that this doesn't appear to be an operational product. Consider removing or rephrasing.

Removed

Figure 1: "Larsen Ice Shelf" should be "Larsen-C Ice Shelf", if it even needs to be labelled at all.

Label has been removed from Fig. 1

Table 2: I'm not a machine learning user, so this may be a stupid question. Should any of the binary classification metrics have units?

Description added

Figure 3: I think this would be clearer without the manually digitised terminus on. Or at least it would be nice to see a version like that in the supplementary information.

We have made the figure easier to read by adding transparency to the markers and changing the order in which they are plotted.

Benjamin Davis

**Anonymous Referee #2, 11 Mar 2024**

**General Comments:**

This work consists of a automatically generated glacial termini data product for 19 key outlet glaciers along the Antarctica Peninsula, and includes 2064 calving front locations from 2013 to 2023 at sub-seasonal temporal resolutions.

The manuscript covers the current state of the art in the field of machine learning/deep learning based cryosphere data extraction methods, as well as the need for such methods to be applied towards glacial data extraction. It describes the importance of calving front data for understanding dynamic glacier changes of marine terminating glaciers, and improving ice modeling. To address the labor-intensive obstacles required by manual delineation, an automatic deep learning-based processing system is developed to extract glacier fronts from satellite imagery. By leveraging the generalization capability of machine learning techniques to provide new observational constraints, this study contributes to groundwork that will enhance the cryosphere community's understanding of glacial dynamics and ice-ocean interactions.

The method uses the deep neural network trained on existing datasets to process Landsat 8 & 9 multiband imagery at spatial resolutions of 30m, and output Shapefile polylines at a spatial accuracy of 59.3 ± 5.9 m (average distance between the measured and predicted fronts). This which falls within human levels of accuracy (<107 m, Goliber et al. 2022).

The dataset itself is composed of zip files of the 19 basins, which are further organized into folders for each observed date, which then contain 2 set of Shapefiles (1 for the coastline, and 1 for the extracted glacial front). Metadata provides the name and date of the processed front. While the scope is small, the dataset still provides valuable new observational constraints.

We have significantly increased the size of our data product. It now includes 42 glaciers (4817 calving front traces) instead of the original 19 glaciers (2064 traces).

The publication is well done, and is largely free of grammatical errors and typographical issues. There are minor remarks to be addressed by the authors, after which I can recommend acceptance at the editor's discretion.

**Specific Comments:**

Dataset Coastline Quality

While the majority of the dataset is well curated, there are some coastlines in the dataset (i.e., drygalski_20210301_coastline, murphy_wilkinson_20191114_coastline, cayley_20141015_coastline, cayley_20200227_coastline) that seem to have erroneous delineations, particularly along the domain boundaries. More validation or pruning of these data is needed, i.e. by manual pruning through visual

GIS software, or some automated pruning by checking inter-annual differences between fronts to detect outliers.

The entire dataset has been reprocessed (see RC1) using the new width-dependent filtering.

Dataset Coastlines Polygons

In conjunction with the above comment, it would be useful to have the glacial termini data in the form of land/ocean polygonal masks in addition to just a polyline, though this may be outside the scope of this work. This would also resolve the errors along the domain boundaries. Alternatively, provision of the domain boundaries would be helpful, as this would make it easier for modelers/community members to judge where errors are, and/or where the coastlines can be stitched to existing land/ocean masks.

Polygon masks are included for all coastline predictions.

Dataset format

The organization of the dataset could be streamlined, such that the user can load an entire time series of a single domain without having to enter/navigate individual folders for each date, and/or make it more manageable for GIS software on less capable machines to load in all at once. Alternatively, such shapefiles could be consolidated, and the ability of Shapefiles to hold multiple features/delineations within a single file would be of use. Provision of monthly, quarterly, annual, or full time series files (similar to IceLines, Baumhoer et al., 2023) should be within scope.

The updated date product now includes consolidated files for each glacier with all calving front traces. We have also included a new annual product for each glacier. All calving fronts in the annual product have been manually validated. We have not included a time series with monthly and quarterly resolution, as this is not possible due to the data gap during the polar night.

Full time series results (area changes) are available for all 42 glaciers (https://doi.org/10.25532/OPARA-557). Details are given in section 4.

Accuracy Comparison w.r.t. Other Datasets

The mean/median distance and binary classification metrics are established accuracy measures in the calving front delineation field, and this study performs well on the evaluated test set. Considering L113P6: ("Although completely different test data sets are involved…Loebel et al. (2023c)."), it may be within scope to see a comparison with existing test sets/studies, to ensure the chosen test set is not biased, and the accuracy metrics are comparable. That being said, the generalization of the network is recognizable, so this can be done at the author's discretion.

Validation results using the CALFIN and ESA-CCI test sets (used in *Loebel et al. 2024*) are included in the revised version. This includes the new Hausdorff distance estimate.
See also the changes to the accuracy assessment in Section 2.

**Minor comments:**

- It would be helpful to provide the spatial accuracy of the data to readers in the abstract.

Accuracy is now included in the abstract.

---

## Author Response (AR2)

Dear Charles Amory,

Thank you for your work on our manuscript. Below you find our responses to the reviewer's questions. We implemented the corresponding technical revisions and clarifications into the manuscript. Please find the revised version together with a track change file.

Thank you and best wishes,

Erik and all co-authors

**Anonymous referee #3, 16 Oct 2024**

The paper by Loebel et al. presents a new dataset of calving front locations, derived from deep learning techniques, in the Antarctic Peninsula. The paper has already gone through thorough rounds of revision and is now almost ready to be published. I have only minor comments on the methodology used here:

(1) The input image size for the ANN is 512x512 pixels. Did you perform a sensitivity analysis of the final results regarding this input data size? What would be the impact of expanding the image size? Is it just more computational time, or would the model also learn more slowly (since there would be more information in a single picture)?

Our input tile size of 512 pixels by 512 pixels, together with the 30 m image resolution of Landsat-8/9 equals about 15 km by 15 km. This value was selected as the majority of Greenland outlet glacier fronts fit into this window (*Loebel et al., 2024*) while still ensuring that we don't have to downsample the input data. This value also works for AP outlet glaciers as all but one glacier (Hektoria-Green-Evans) fit in this input window.

Increasing the input tile size further would therefore not result in more ice-ocean boundary, but would simply include more land and ocean area. This would cause our reference dataset to lean more towards the land-ocean boundary and less towards the ice-ocean boundary. As the ANN model is trained to segment the entire input tile, it would be optimized more towards the land-ocean boundary than the ice-ocean boundary, which is not what we want for calving front extraction. In addition, the computational cost will increase.

In the current version of the manuscript the values of 512 pixels by 512 pixels seem a bit arbitrary. We thank the reviewer for this question and include an explanation in section 2.1 (P4 L81).

(2) The training dataset used is from the Greenland ice sheet. I believe this was mainly from Landsat 8-9. Other studies (Wood et al., 2021; Cheng et al., with Calfin) produced large-scale ice front datasets in Greenland from Landsat 4-8. I am wondering why the model was not trained with all these different types of sensors to provide longer time coverage?

Our processing system is built around maximizing the impact of the multispectral capabilities (these lead to more accurate calving front extractions (*Loebel et al. 2022*)) and higher image acquisition rate of Landsat-8/9. As the older Landsat satellites have different multispectral channels and less coverage across the spectrum, our model cannot be applied directly. It would require significant changes to reference data and retraining. In addition, a major motivation for automated calving front delineation is the higher image acquisition rate of modern satellite constellations.

However, we agree that there would be significant value in applying this or a similar approach to older satellite data, as has been done for the CALFIN product (*Cheng el al., 2021*) for Greenland. Such a more historical machine delineated calving front data record does not yet exist for the AP. We gladly include this in our manuscript's conclusions (P14 L270).

(3) Regarding the training dataset, did you provide the model with some cloudy data? This could help the model learn when not to delineate fronts.

Partially cloudy satellite images (where the calving front is not completely obscured) are included in both the training and test datasets. This is important to maximize the extraction rate for our processing system and to increase the sampling of the data product. This is already mentioned in the manuscript in section 2.1 (P5 L91). An example is also shown in Figure 3 (f).

Concerning completely clouded images (calving front not visible): Our processing system is built around classifying only between land/glacier and ocean pixels. Therefore, cloud pixels are classified as land/glacier or ocean depending on their spatial context. If there is no spatial context, pixels will still be classified as land/glacier or ocean, but with large uncertainties or even random in extreme cases. Therefore, there is no way to directly (i.e. within the model prediction) discard an extraction based on cloud cover. In our processing system, completely clouded images result in failed or incorrect calving front delineations, which are identified and discarded during post-processing.

(4) Did you compare your results in the Antarctic Peninsula with previous datasets made from MODIS imagery (https://tc.copernicus.org/articles/17/2059/2023/tc-17-2059-2023.pdf)? Additionally, there is no mention of the Green et al. dataset, which delineated calving fronts all around Antarctica using multi-sensor data (see here: https://www.nature.com/articles/s41586-022-05037-w).

The MODIS dataset (*Andreasen et al., 2023*) as well as the dataset by *Greene at al. (2022)* are concerning circum-Antarctic ice shelf calving front locations. For the AP they include the Larsen Ice Shelf but not the outlet glaciers which are the focus of our manuscript. They are comparable to the machine generated IceLines dataset (*Baumhoer et al., 2023*) which we mentioned in the introduction as a motivation for our work (as it also does not include AP outlet glaciers). Our results are therefore not directly comparable to these datasets.

To make this clearer in the manuscript, we have made some minor changes in Section 1 and expanded the description of Table 1 to better distinguish AP glacier fronts from ice shelf fronts (*Andreasen et al., 2023* and *Greene at al. (2022)* are now also mentioned).

Overall, I think the figures are in good shape, and I recommend the paper for publication after minor technical revisions.

**References:**

Andreasen, J. R., Hogg, A. E., and Selley, H. L.: Change in Antarctic ice shelf area from 2009 to 2019, The Cryosphere, 17, 2059–2072,https://doi.org/10.5194/tc-17-2059-2023, 2023.

Baumhoer, C. A., Dietz, A. J., Heidler, K., and Kuenzer, C.: IceLines–A new data set of Antarctic ice shelf front positions, Scientific Data, 10, 138, 2023.

Cheng, D., Hayes, W., Larour, E., Mohajerani, Y., Wood, M., Velicogna, I., and Rignot, E.: Calving Front Machine (CALFIN): Glacial Termini Dataset and Automated Deep Learning Extraction Method for Greenland, 1972–2019 , The Cryosphere, 15, https://doi.org/10.5194/tc-15-1663-2021, 2021

Loebel, E., Scheinert, M., Horwath, M., Heidler, K., Christmann, J., Phan, L. D., Humbert, A., and Zhu, X. X.: Extracting Glacier Calving Fronts by Deep Learning: The Benefit of Multispectral, Topographic, and Textural Input Features, IEEE Transactions on Geoscience and Remote Sensing, 60, 1–12, https://doi.org/10.1109/TGRS.2022.3208454, 2022.

Loebel, E., Scheinert, M., Horwath, M., Humbert, A., Sohn, J., Heidler, K., Liebezeit, C., and Zhu, X. X.: Calving front monitoring at a subseasonal resolution: a deep learning application for Greenland glaciers, The Cryosphere, 18, 3315–3332, https://doi.org/10.5194/tc-18-3315-2024, 2024.

Greene, C. A., Gardner, A. S., Schlegel, N.-J., and Fraser, A. D.: Antarctic calving loss rivals ice-shelf thinning, Nature, 609, 948–953,350 https://doi.org/http://dx.doi.org/10.1038/s41586-022-05037-w, 2022.